# CONTROLLING THE MEMORABILITY OF REAL AND UN-REAL FACE IMAGES

## ABSTRACT

Everyday, we are bombarded with many photographs of faces, whether on social media, television, or smartphones. From an evolutionary perspective, faces are intended to be remembered, mainly due to survival and personal relevance. However, all these faces do not have the equal opportunity to stick in our minds. It has been shown that memorability is an intrinsic feature of an image but yet, it is largely unknown what attributes make an image more memorable. In this work, we aimed to address this question by proposing a fast approach to modify and control the memorability of face images. In our proposed method, we first found a hyperplane in the latent space of StyleGAN to separate high and low memorable images. We then modified the image memorability (while maintaining the identity and other facial features such as age, emotion, etc.) by moving in the positive or negative direction of this hyperplane normal vector. We further analyzed how different layers of the StyleGAN augmented latent space contribute to face memorability. These analyses showed how each individual face attribute makes an image more or less memorable. Most importantly, we evaluated our proposed method for both real and unreal (generated) face images. The proposed method successfully modifies and controls the memorability of real human faces as well as unreal (generated) faces. Our proposed method can be employed in photograph editing applications for social media, learning aids, or advertisement purposes.

## 1 INTRODUCTION

In our everyday life, we are exposed to many pictures of scenes, objects and faces. Research has shown that all images do not have the same likelihood to be recalled later (Isola et al., 2011; Bainbridge et al., 2013; Isola et al., 2014). Although different people have different abilities in memorizing visual contents (image or video), it has been shown that memorability is an intrinsic feature of an image and it is consistent across different observers (Isola et al., 2011; Bainbridge et al., 2013; Isola et al., 2014). In other words, memorability of an image is an attribute of that image which can be measured, predicted or manipulated (Isola et al., 2011). So far there have been several studies that have attempted to understand, predict, and even modify image memorability. (Khosla et al., 2013a; 2015; Goetschalckx & Wagemans, 2019; Fajtl et al.; Needell & Bainbridge, 2021; Squalli-Houssaini et al., 2018; Almog et al.), and a few work attempted to modify image memorability (Khosla et al., 2013b; Siarohin et al., 2017; Sidorov, 2019; Goetschalckx et al., 2019). However, for practical applications (e.g., education and advertisement), it is most important to have methods for memorability modifications. Moreover, such approaches will help understanding what constitute as image memorability, i.e. "What makes an image memorable?".

This work proposes a new framework based on Generative Adversarial Networks (GANs) (Goodfellow et al., 2014) for modifying face memorability as a facial attribute. Older approaches on modifying face memorability manually used face features (Khosla et al., 2013b), such as SIFT (Lowe, 2004), HOG2x2 (Dalal & Triggs, 2005), and Local Binary Pattern (LBP) (Ojala et al., 2002). Recently, GANs have been used to modify the memorability of images. (Goetschalckx et al., 2019; Sidorov, 2019). Goetschalckx et al. (2019) leverage latent vector modification to change the memorability of the fake food, scenes, and animal images generated by BigGAN (Brock et al., 2018). This memorability modification affects several attributes of the image, such as size, color, and shape. In our work, we aim to modify memorability of face images of real people while keeping their identity, consequently, their method cannot be used here.

The largest dataset of the human faces with their memorability annotations is US 10k Face database (Bainbridge et al., 2013), which includes 2222 face images with their memorability scores acquired from human observers in an experiment. StyleGANs are the state-of-the-art models for generating real-looking faces. Our utilization of StyleGANs is required to create a dataset of realistic-looking faces. Not only that, for modifying the memorability of real faces, we need Style-GANs to reconstruct real faces with high accuracy. To date, StyleGANs are the state-of-the-art models in reconstructing real-face images. StyleGANs provide an extended latent space which we leverage to derive a more accurate memorability hyperplane. Also, the face attributes of Style-GANs are especially disentangled in comparison to other GANs, which is required to accurately modify faces for memorability and study the attributes contributing to this. For this, we employed pre-trained StyleGAN1 (Karras et al., 2019) and StyleGAN2 (Karras et al., 2020) on the FFHQ dataset (Karras et al., 2019) to generate 100k fake faces. Next, we adopted computational memorability models which we trained on the US 10k Face Database, to predict the memorability of the generated face images and organized them into faces with high or low memorability. Inspired by Shen et al. (2020), we found a hyperplane in latent subspace to separate the highly-memorable and low-memorable faces. We showed that both latent space and extended latent space can be used for finding the separating hyperplane. After finding the hyperplane, we moved the latent vector of each image, in the positive or negative direction of the normal vector of that hyperplane and changed the distance of the latent vector from the separating hyperplane to manipulate the memorability of that image. We name the normal vector of this hyperplane, memorability modification vector. With this proposed approach, we could control the amount of change in memorability by using different weights for the memorability modification vector. In contrast to the method proposed by Sidorov (2019), our method does not require training another auxiliary network for modifying face memorability and the amount of change in memorability is controlled by a hyperparameter.

Since different hyperplanes for different facial attributes in StyleGAN latent space (Shen et al., 2020; Härkönen et al., 2020) can be found, our method can be used to modify the memorability of the images conditionally. For example, we are able to change the memorability of the face while maintaining the length of the hair and the existence of eyeglasses. For this, we first find the corresponding hyperplanes for these attributes, and then leverage projection to those subspaces to have the desired attributes fixed while changing memorability. StyleGAN produces high quality real-looking images which are near impossible to differentiate from real images. To make sure our memorability-modified faces still look real, we considered the Frechet Inception Distance (FID) (Heusel et al., 2017) and Kernel Inception Distance (KID) (Bińkowski et al., 2018) scores of the generated faces from the StyleGAN as the baseline. Then by calculating the FID score and KID score of our memorability modified images, we showed that our modified faces still looked real.

For real faces, we first embedded the face images into GAN latent space using Image2Style (Abdal et al., 2019) and the method provided by Karras et al. (2020). After finding the image latent vector, we modified the real face memorability in the same way previously explained for synthetic faces. Figure 1 illustrates the general idea of our approach. Finally, we examined how different layers of extended latent space of StyleGAN affect the image of each face and its memorability.

## 1.1 RELATED WORKS

**Image Memorability.** People have different capabilities in memorizing different visual events (Hunt et al., 1981). In spite of these differences, through a series of experiments, Isola et al. (2011) showed that people consistently remember some images and forget others. They designed an online memory game experiment and recruited a large number of participants through Amazon Mechanical Turk. In this experiment, participants observed a series of images presented in a sequence and were tasked to detect repetitive images in the sequence. Then Isola et al. (2011) measured a memorability score for each image, which corresponded to the rate of people remembering an image after single exposure to that image in the sequence. Khosla et al. (2015) created the largest annotated image memorability dataset (LaMem), which consists of 60,000 images, mostly objects, scenes, and animals. Using this dataset, they introduced the first deep model for predicting image memorability. This model uses AlexNet (Krizhevsky et al., 2012) as its backbone architecture. Moreover, Needell & Bainbridge (2021) introduced new architectures based on residual networks (He et al., 2016) to improve the performance of memorability prediction. In addition to these models, Fajtl et al. leveraged Attention Maps to introduce AMNet for predicting memorability.

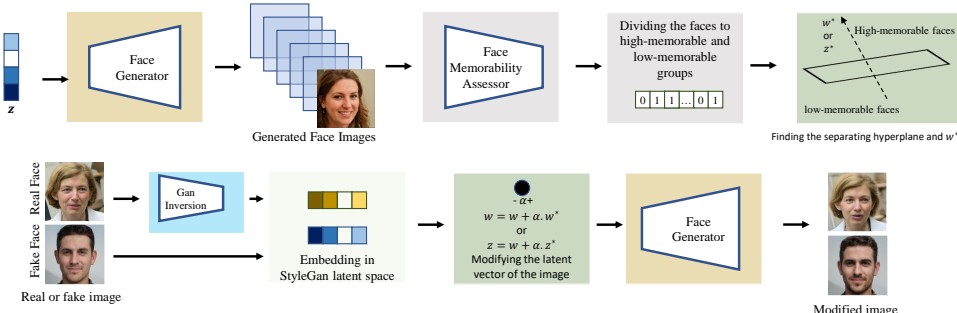

Figure 1: **The proposed method.** We first generate synthesized face images from random latent vectors and predict their memorability scores by a face memorability assessor network. Then, divide them into high-memorable and low-memorable faces. Using either latent vector or extended latent vector subspace we find a memorability separating hyperplane in that subspace. The second row shows the proposed framework for modifying the memorability score of face images. Using a GAN inversion technique we first map faces to StyleGAN's latent space. Then modify the face memorability by moving the latent vector (or extended latent vector) towards negative or positive direction of memorability discriminating hyperplane obtained in previous step. The modified latent vector (or extended latent vector) is fed to the GAN to generate the face image with modified memorability. It is optional to ovalize the face images before feeding to the assessor.

**Generative Adversarial Networks (GANs).** With the development of Generative Adversarial Networks (Goodfellow et al., 2014), we are now capable of generating real-looking synthetic images that are indistinguishable from real images. Generally, these networks are composed of two parts; a generative and a discriminative network. The goal of the generative network is to generate real-looking images to fool the discriminator and the goal of the discriminator is to learn to distinguish generated images from real images. These two networks are optimized through a minmax game where both sides compete to reach their specified goals. In recent years, there have been huge improvements in this area and many different GANs have been introduced to produce natural-looking images such as Progressive GAN (Karras et al., 2017), DCGAN (Radford et al., 2015), and Cycle-GAN (Zhu et al., 2017). In this work, we used pre-trained StyleGAN1 (Karras et al., 2019) and StyleGAN2 (Karras et al., 2020) on the FFHQ dataset (Karras et al., 2019) to generate real-looking faces.

**Modifying Image Memorability.** While image memorability modification has many potential applications (e.g. in education or advertisement), it has not been adequately investigated. Khosla et al. (2013b) proposed a pioneering method for changing face memorability. It leveraged Active Appearance Models (AAMs (Cootes et al., 2001)) to represent faces by their shape and appearance. Then the loss function was defined based on the cost of modifying identity, modifying facial attributes, and memorability. As a result, in their method, the identity was fixed. Another approach (Sidorov, 2019) used VAE/GAN (Larsen et al., 2016), StarGAN (Choi et al., 2018), and AttGAN (He et al., 2019) and trained them with three memorability levels (poorly memorable, moderately memorable, and highly memorable) of faces and modified the memorability of different faces to these three levels only. Additionally, Siarohin et al. (2017) utilized style transfer to increase the memorability of an input image. However, the added style adversely affected the realness of the modified images, such that it barely could be used in real-world applications. Most recently, Goetschalckx et al. (2019) trained a transformer network to change the memorability of each generated image by Big-GAN (Brock et al., 2018) through modifying their latent vectors. Their method works on generated images of objects and scenes.

## 2 METHOD

The overview of our proposed method is depicted schematically in Figure 1. Below we explain the approach step by step.

## 2.1 CREATING THE DATASET

For the purpose of analyzing the latent vectors of the GANs and their relation to memorability, we need a large dataset of face images with their memorability scores. The largest dataset available for face images is the 10k US Adult Faces Database (Bainbridge et al., 2013). This database contains 10,168 natural human face images and 2,222 of these images are annotated with memorability scores. To create a larger dataset for face images with their corresponding memorability scores, we leveraged StyleGAN1 and StyleGAN2, which are the state-of-the-art models for creating realistic-looking face images. These models were pre-trained on the FFHQ dataset (Karras et al., 2019) which consists of 70,000 high-quality face images with $1024 \times 1024$ resolution with variations in age, gender, and glasses. We created two different datasets with StyleGAN1 and StyleGAN2. We randomly sampled 100k $z \in \mathbb{R}^{1 \times 512}$ from a standard normal distribution with truncation to produce high-quality synthetic face images and saved their mappings in the extended latent space ($\mathbb{R}^{18 \times 512}$) of both GANs.

## 2.2 PREPROCESSING STEP

The generated images from the StyleGAN have $1024 \times 1024$ resolution in three channels. For the purpose of acquiring the memorability of the generated images, we had to preprocess them to predict their memorability with the computational memorability predicting models (assessor). We leveraged VGG16 (Simonyan & Zisserman, 2014), ResNet50 (He et al., 2016), and SENet50 (Hu et al., 2018) that are pre-trained on VGGFace2 (Cao et al., 2018) and fine-tuned them on the US 10k Face Database to correctly estimate face memorability scores. This dataset consists of oval-shaped human faces with white backgrounds. 10k US Adult Face images have the same height of 256 with different widths. Hence, we had to first ovalize the generated faces, then compute their memorability scores. We used MTCNN (Zhang et al., 2016) for detecting the face in the generated images and masked an oval on it to make all the images similar to the US 10k Face Database (See Figure 2).

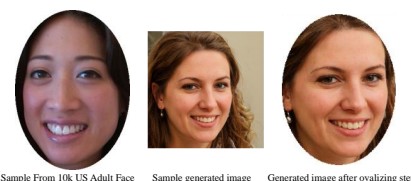

Sample From 10k US Adult Face    Sample generated image    Generated image after ovalizing step

Figure 2: Preprocessing step to make the shape of the synthesized images similar to the dataset that the assessor is trained on.

## 2.3 PREDICTING THE MEMORABILITY SCORES

As explained in previous section, memorability assessors are trained on 10K US face database which includes ovalized faces. We first, tested whether the ovalization process is necessary to obtain memorability scores for generated faces. For this, we fine-tuned the pre-trained SENet50 (Hu et al., 2018), VGG16 (Simonyan & Zisserman, 2014), and ResNet50 (He et al., 2016) for predicting face memorability scores on 10K US face database. We calculated the memorability of both the oval-shaped and square-shaped faces generated from StyleGAN1 and StyleGAN2. The distributions of the memorability scores using SENet with oval and squared faces are shown in Figure 3. As can be seen, the memorability distributions are very similar for oval and square faces. We further calculated the Kendall rank correlation (Schaeffer & Levitt, 1956) and Spearman's rank correlation for the paired oval-shaped and square-shaped images. We observed that SeNet50 and VGG16 have a high-rank score (see Table 1). These results showed that these models can also be used for the square-shaped face images that contain a background. The benefit of using square-shaped faces is that we can have more control on the hair of the person. Please note that the performance of memorability predicting models are measured with rank correlation (Khosla et al., 2015).

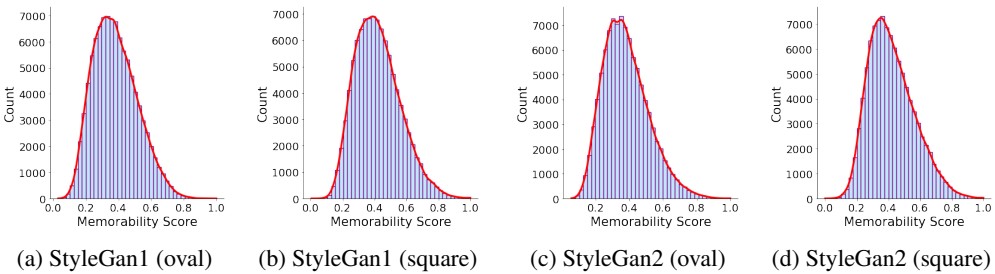

| (a) StyleGan1 (oval) | (b) StyleGan1 (square) | (c) StyleGan2 (oval) | (d) StyleGan2 (square) |

Figure 3: Distributions of the memorability scores of the generated images of StyleGan1 and Style-Gan2 based on using an ovalization step or not.

Table 1: Correlations of memorability scores of square-shaped and oval-shaped faces. High correlation score suggests the assessor performs well on square faces too.

| Assessor | Kendall Tau correlation | Spearman's correlation |
|---|---|---|
| ResNet50 | 0.1606 | 0.2391 |
| SENet50 | 0.4419 | 0.6217 |
| VGG16 | 0.4720 | 0.6562 |

## 2.4 THE PROPOSED METHOD FOR MODIFYING FACE MEMORABILITY

Next we aimed to find a hyperplane to separate the highly-memorable and low-memorable images. First, we needed to label the faces into highly-memorable and low-memorable faces. For this, we used the mean of memorability scores as a threshold. We labeled an image as highly or low memorable if its memorability score was higher or lower than the mean. In our experiments, we also tried using median of the memorability scores as the threshold for labeling high and low memorable groups. Next, using logistic regression, we attempted to find a hyperplane to separate low-memorable and highly-memorable images. For this purpose, we used either $\boldsymbol{z} \in \mathbb{R}^{1 \times 512}$ latent vectors or $\boldsymbol{w} \in \mathbb{R}^{18 \times 512}$ extended latent space of StyleGAN1 and StyleGAN2. After finding the separating hyperplane, we moved the latent vector or the augmented latent vector in the positive or negative direction of the normal vector of that hyperplane to control the image memorability. This hyperplane denotes the moderately memorable images, as we chose it to separate the faces into high-memorable and low-memorable faces based on the mean (or median) of the memorability scores. Memorability of each image is related to the distance of its latent vector (or extended latent vector if extended latent vector is used to find the hyperplane) from this hyperplane. Consider image $i$ with corresponding latent vector $\boldsymbol{z}_i \in \mathbb{R}^{1 \times 512}$ or $\boldsymbol{w}_i \in \mathbb{R}^{18 \times 512}$, and its memorability score $mem_i$. We note the normal vector of this hyperplane by $\boldsymbol{z}^* \in \mathbb{R}^{1 \times 512}$ or $\boldsymbol{w}^* \in \mathbb{R}^{18 \times 512}$ and we show the distance from the hyperplane by function $d$. We will have:

$$mem_i \propto d(\boldsymbol{z}^*, \boldsymbol{z}_i) = \boldsymbol{z}^{*^T}.\boldsymbol{z}_i \tag{1}$$

$$mem_i \propto d(\boldsymbol{w}^*, \boldsymbol{w}_i) = \boldsymbol{w}^{*^T}.\boldsymbol{w}_i \tag{2}$$

Hence, we can change the memorability of each image, by changing the distance of its latent vector (or extended latent vector) from the separating hyperplane; $\boldsymbol{z}_{edit} = \boldsymbol{z} + \alpha \boldsymbol{z}^*$ or similarly $\boldsymbol{w}_{edit} = \boldsymbol{w} + \alpha \boldsymbol{w}^*$ and we will have:

$$d(\boldsymbol{z}^*, \boldsymbol{z}_{edit}) = \boldsymbol{z}^{*^T}.\boldsymbol{z}_{edit} = \boldsymbol{z}^{*^T}.(\boldsymbol{z} + \alpha \boldsymbol{z}^*) = \boldsymbol{z}^{*^T}.\boldsymbol{z} + \alpha = d(\boldsymbol{z}^*, \boldsymbol{z}) + \alpha \tag{3}$$

$$d(\boldsymbol{w}^*, \boldsymbol{w}_{edit}) = \boldsymbol{w}^{*^T}.\boldsymbol{w}_{edit} = \boldsymbol{w}^{*^T}.(\boldsymbol{w} + \alpha \boldsymbol{w}^*) = \boldsymbol{w}^{*^T}.\boldsymbol{w} + \alpha = d(\boldsymbol{w}^*, \boldsymbol{w}) + \alpha \tag{4}$$

The results of this classification task for finding the separating hyperplane are presented in Table 4 when different assessors are used for memorability score predictions. In each case, the accuracy was about 10 percent higher when we used extended latent space ($w$), hence we showed the results for extended latent space in Table 4. According to the results, we decided to use SENet50 as our assessor and mean memorability as the threshold for our further analysis.

Table 2: Accuracy of the separating hyperplane, based on the method for dividing images into highly-memorable and low-memorable images, the shape of the images, and the assessor.

| Assessor | Median | | Mean | |
|---|---|---|---|---|
| | Oval | Square | Oval | Square |
| ResNet50 | 0.8131 | 0.7933 | 0.8149 | 0.7928 |
| SENet50 | 0.8157 | 0.8291 | **0.8207** | **0.8317** |
| VGG16 | 0.7938 | 0.8037 | 0.7952 | 0.8071 |

The performance of logistic regression is higher when the extended latent space is used to find the separating hyperplane. Further, as we show in the next section working with the extended latent space yields better results in modifying face memorability. (See A.4 for the latent space results.)

One of the benefits of our approach is that we can fix specific attributes while changing memorability. It has been shown that latent vector play an important role in determining different attributes of a face and we can find hyperplanes to separate faces based on those attributes, such as glasses, age, and smile. Consider the norm vectors of these hyperplanes as $\mathbb{A} = \{\boldsymbol{a}_1, \boldsymbol{a}_2, ..., \boldsymbol{a}_k\}$. We can fix these attributes by subspace projection as follows: $\boldsymbol{w}^*_{new} = \boldsymbol{w}^* - \sum_{i=1}^{i=k}(\boldsymbol{w}^{*T}.\boldsymbol{a}_i)\boldsymbol{a}_i$ (See A.5).

## 2.5 LATENT VECTOR RECOVERY FOR REAL FACES

The efficiency of our method to modify real human faces depends on how well the latent vector of the real face can be obtained to reconstruct the original image. After acquiring the latent vector of the real face image, we can repeat the process for the synthesized images and modify their memorability. In this work, we used image2style (Abdal et al., 2019) to embed the real faces to latent space of StyleGAN1, whose loss function consists of a VGG-16 perceptual loss (Johnson et al., 2016) and a pixel-wise MSE loss term. Furthermore, for projecting real faces to StyleGAN2 latent space, we employed the same algorithm described by Karras et al. (2020) after using the dlib library (King, 2009) to align 68 face landmarks in the preprocessing step. In this work, we assume that we have the latent vector of the generated faces, however, if the latent vector of a generated face is not available, a similar approach can be employed to retrieve the latent vector.

## 3 EXPERIMENTS

### 3.1 CHANGING MEMORABILITY OF SYNTHESIZED FACES

As described in Section 2.1, we created two different datasets with StyleGAN1 and StyleGAN2. We randomly sampled 100k $\boldsymbol{z} \in \mathbb{R}^{1\times512}$ from a standard normal distribution with truncation to produce high-quality synthetic face images and saved their mappings in $\mathbb{R}^{18\times512}$ augmented latent space of both GANs. We calculated their memorability scores using SENet50, then labeled them highly or low-memorable images when compared to the mean memorability score in the dataset, finally we used the logistic regression to find the separating hyperplane as described in Section 2.4. After finding the separating hyperplane, we evaluated our method in modifying the memorability of generated faces with StyleGAN1 and StyleGAN2 using our proposed method. As we discussed earlier in 2.4, we found separating hyperplanes in either latent space or extended latent space of StyleGAN1 and StyleGAN2. Figure 4 and Figure 5, demonstrates some samples of memorability modification with StyleGAN1 and StyleGAN2 respectively (See A.1 for more examples).We observe that increasing the memorability scores causes some decreases in the facial weight, increases the presence of makeup and thickness of the lips, and makes the person look younger. Moreover, it will affect the skin tone by making it brighter. It makes the face more serious. However, decreasing the memorability score has opposite effects.

In addition to using our method to evaluate face images qualitatively, we also tested the performance of our method in modifying memorability, quantitatively. For this, we used 10k synthesised faces and tried different weights of memorability modification on them. Figure 6 depicts the distributions of memorability scores when shifted by our method as well as the mean of the distributions, which suggests us we were successful in modifying the memorability scores of the faces.

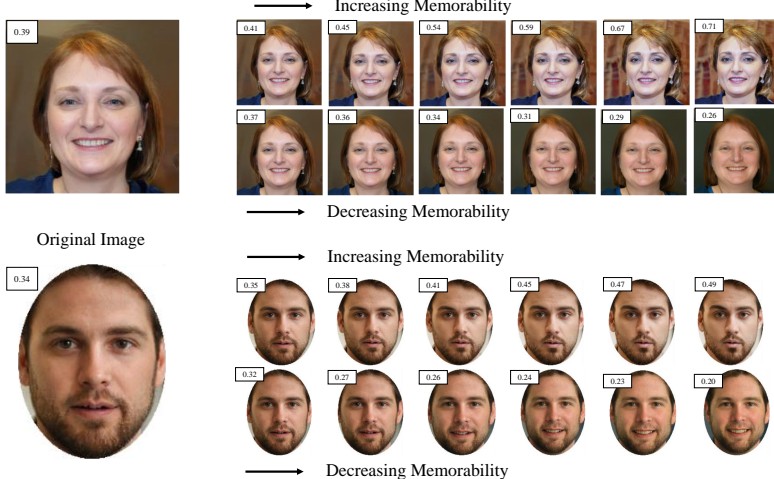

Figure 4: **Modifying memorability of faces generated by StyleGAN1.** Exemplar faces and their counterparts when our memorability modifying approach is applied to them. The second row depicts images when an extra step of ovalization is applied before feeding to the assessor. The corresponding memorability score is presented in the top left corner of each image.

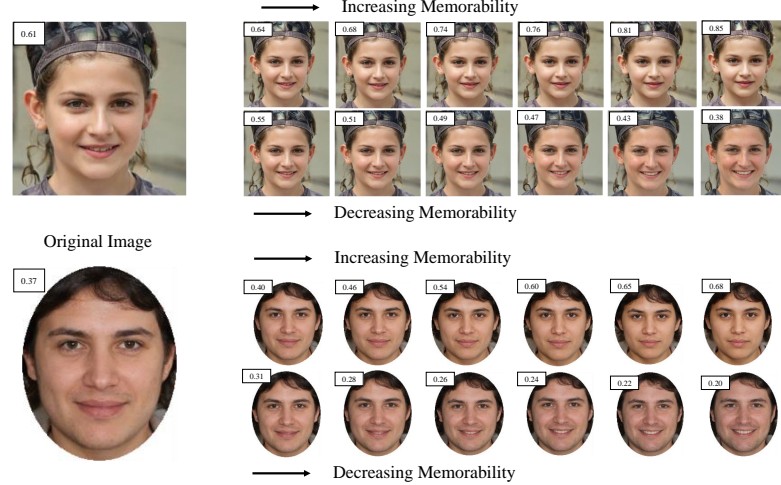

Figure 5: **Modifying memorability of faces generated by StyleGAN2.** Exemplar faces and their counterparts when our memorability modifying approach is applied to them. The second row depicts images when an extra step of ovalization is applied before feeding to the assessor.

## 3.2 REALNESS OF THE MODIFIED FACES

In this experiment, we evaluated the realness of the faces when our memorability modification was applied to them. StyleGAN is a state-of-the-art model in generating real-looking faces. Hence, we considered the realness of the generated images from StyleGAN (before memorability modification) as our baseline and compared them to the modified faces. First we generated 10k synthesized faces, then appllied different weights of memorability modification vector to their latent vectors. We utilized two well-known measures for this purpose; FID and KID. As demonstrated in Figure 7, the FID and KID scores of the modified images are close to the unmodified faces and therefore, the realness of faces is not affected by our method.

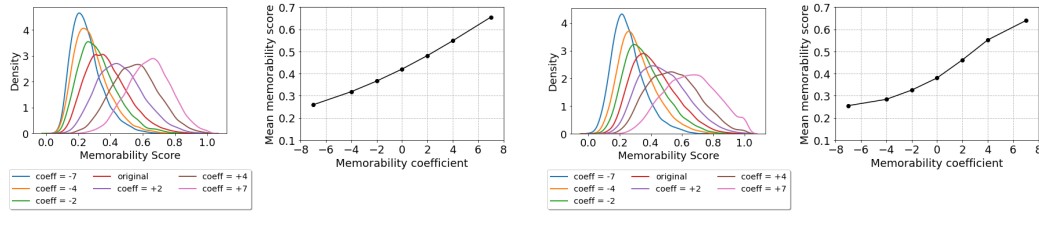

Oval-shape faces                    Square-shape faces

Figure 6: The effectiveness of our method for modifying memorability scores tested on 10k generated faces. As depicted the distribution and mean memorability score of images changes with the coefficient used for memorability modification.

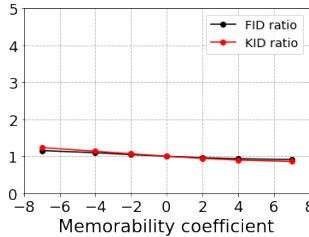

Figure 7: The realness of the 10k generated images measured by the FID and KID ratios for different memorability modification coefficients. FID and KID score of the unmodified images are the respective baselines. A close to one ratio indicates similar level of realness with the baseline.

## 3.3 CHANGING MEMORABILITY OF REAL FACES

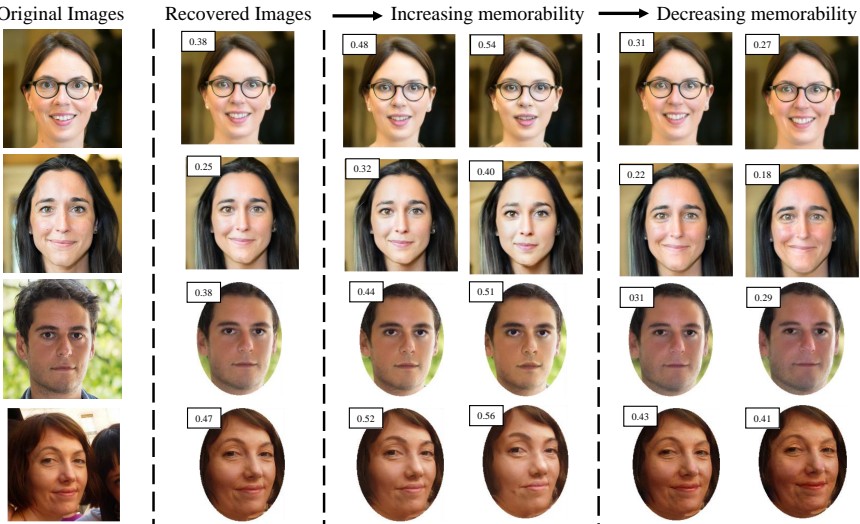

Figure 8: **Memorability modification of real faces.** The first three rows were encoded to Style-GAN2 latent space and the fourth row was encoded to StyleGAN1 latent space.

Next we evaluated our method on real human faces. First, we computed their extended latent vector using the GAN inversion method previously described and then applied our proposed method to them to change their memorability. For real human faces, we chose to work with the extended

latent vectors, because the regenerated images from the extended latent space are more similar to the original images (See Figure 8).

### 3.4 LAYERWISE MEMORABILITY MODIFICATION

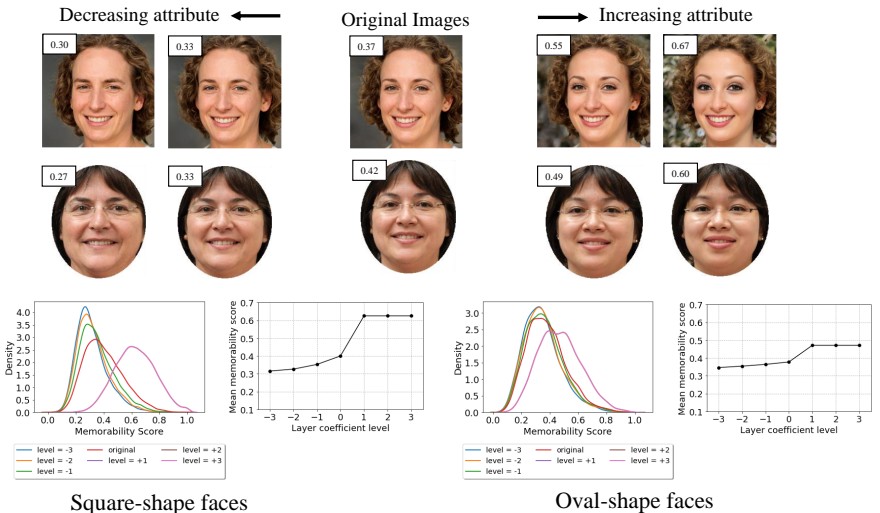

Figure 9: **The effect of changes in the 7th layer of the extended latent space on memorability score.** This layer mostly affects the presence of makeup in female and facial hair in male faces. We observe that when we move this layer in the positive direction of the corresponding layer of the memorability vector ($w^*$), the memorability score will drastically increase and the presence of makeup is highlighted. Additionally, we observed that the lips thickness increases, and the eyebrows shape and the skin tone of the person changes changes.

Next we tried to identify which layers in extended latent vectors contributed the most to face memorability, i.e., which layers are most responsible for modifying a face memorability score. In each experiment, we only changed one layer and kept other layers the same. We plotted the faces after these layerwise changes to examine what kind of changes these layerwise modifications caused in the faces. We observed that modifications in the first 11 layers were mostly responsible for changes in a face, and the other 7 layers mostly just affected the color and background of the image. Hence, we only focused on the first 11 layers. We then repeated the same layerwise changes on 3k synthesized faces and calculated the mean memorability of these images before layerwise and after layerwise modifications (See Figure 9, for the other layers see A.2).

## 4 CONCLUSION

In this work, we proposed a new method for modifying the memorability of face images. Our approach does not suffer from the limitations of the previous methods and is able to modify the memorability of faces (synthetic and real) within an arbitrary continuous range. Moreover, we demonstrated these changes will not affect the realness of the images. However, if a very large weight for the memorability modification vector is chosen, it will affect the face identity and realness. We showed that our method is effective by applying it to 10k synthesized faces. Further, we employed our method on real human faces and showed it is effective in changing the memorability of real faces as well. Finally, we studied how layerwise modifications will affect the face and its memorability score and discussed one of the benefits of the proposed method is modifying the memorability score conditionally by leveraging subspace projection method.

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

# A  APPENDIX

## A.1  ADDITIONAL EXAMPLES

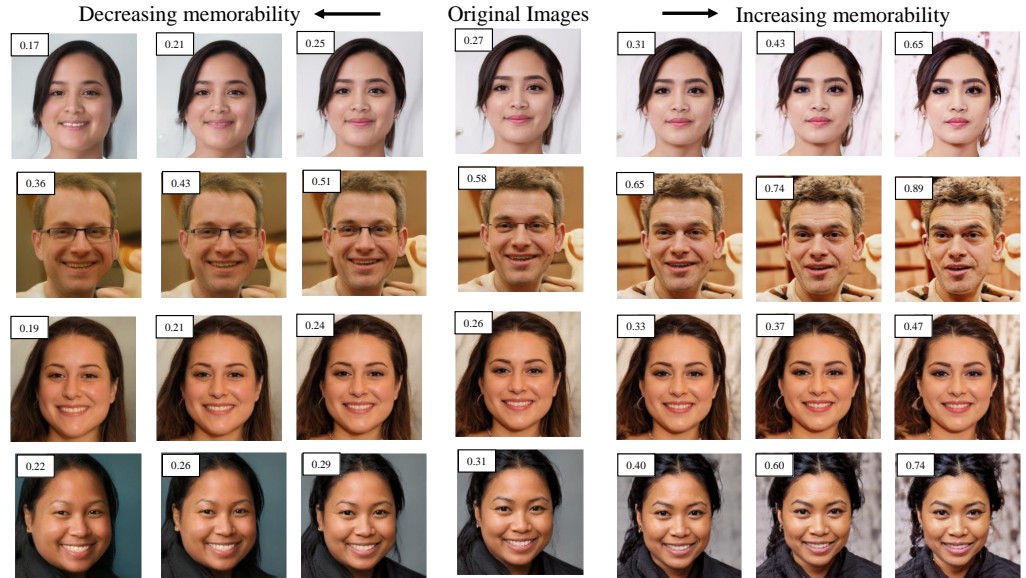

Figure 10: **Generated images by StyleGAN1 with their corresponding memorability scores.** Modified images when extended latent space of the StyleGAN1 was used to determine the separating hyperplane. Square-shaped faces were fed to the assessor.

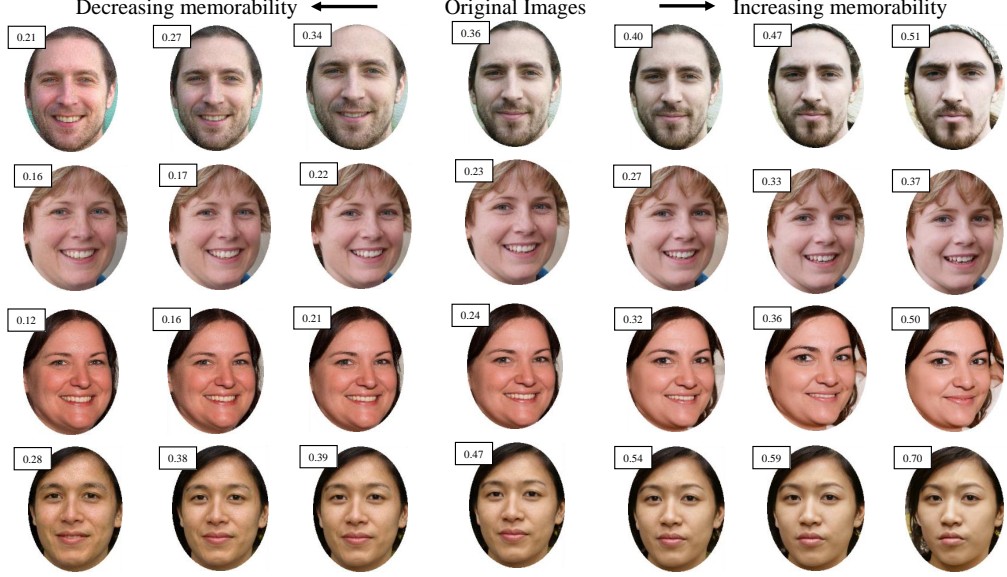

Figure 11: **Generated images by StyleGAN1 with their corresponding memorability scores.** Modified images when extended latent space of the StyleGAN1 was used to determine the separating hyperplane. Oval-shaped faces were fed to the assessor.

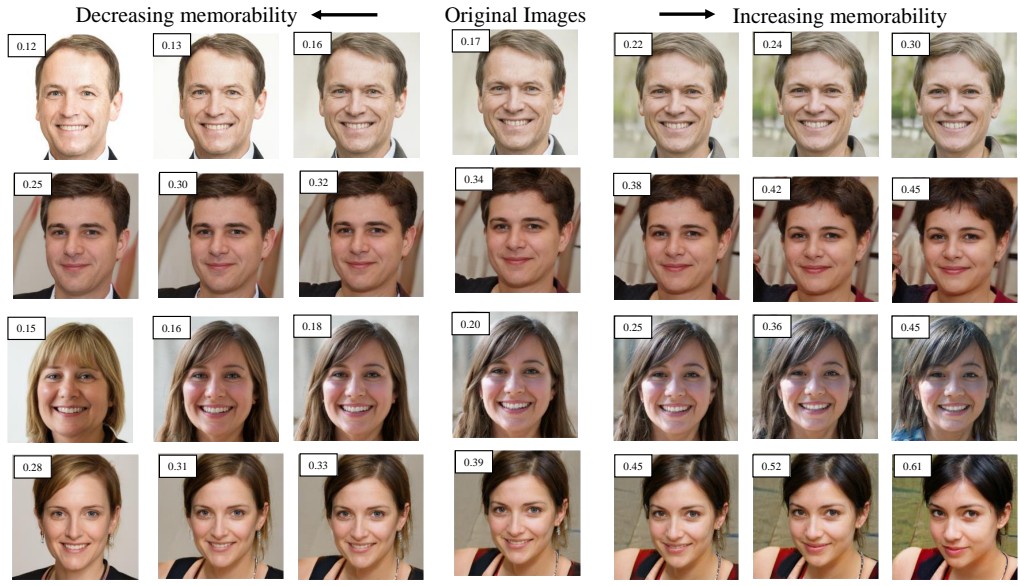

Figure 12: **Generated images by StyleGAN1 with their corresponding memorability scores.** Modified images when latent space of the StyleGAN1 was used to determine the separating hyperplane. Square-shaped faces were fed to the assessor.

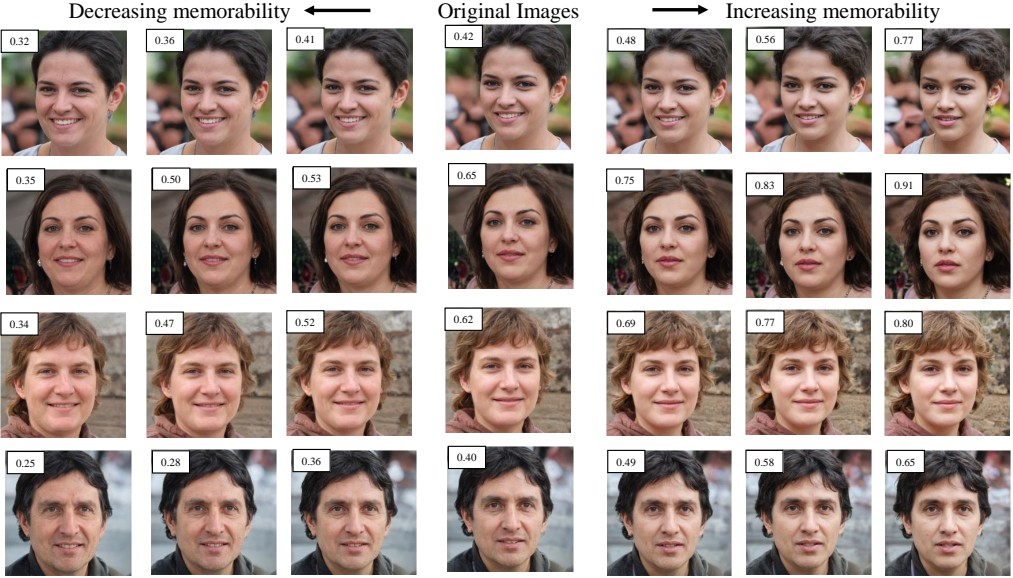

Figure 13: **Generated images by StyleGAN2 with their corresponding memorability scores.** This figure shows the modified images when extended latent space of the StyleGAN2 was used to determine the separating hyperplane. Square-shaped faces were fed to the assessor.

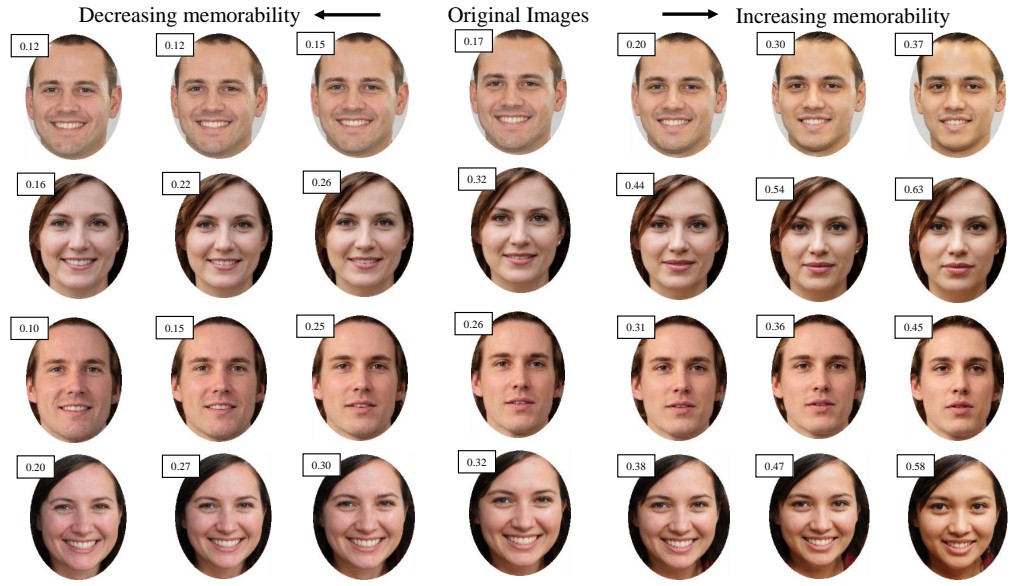

Figure 14: **Generated images by StyleGAN2 with their corresponding memorability scores.** This figure shows the modified images when extended latent space of the StyleGAN2 was used to determine the separating hyperplane. Oval-shaped faces were fed to the assessor.

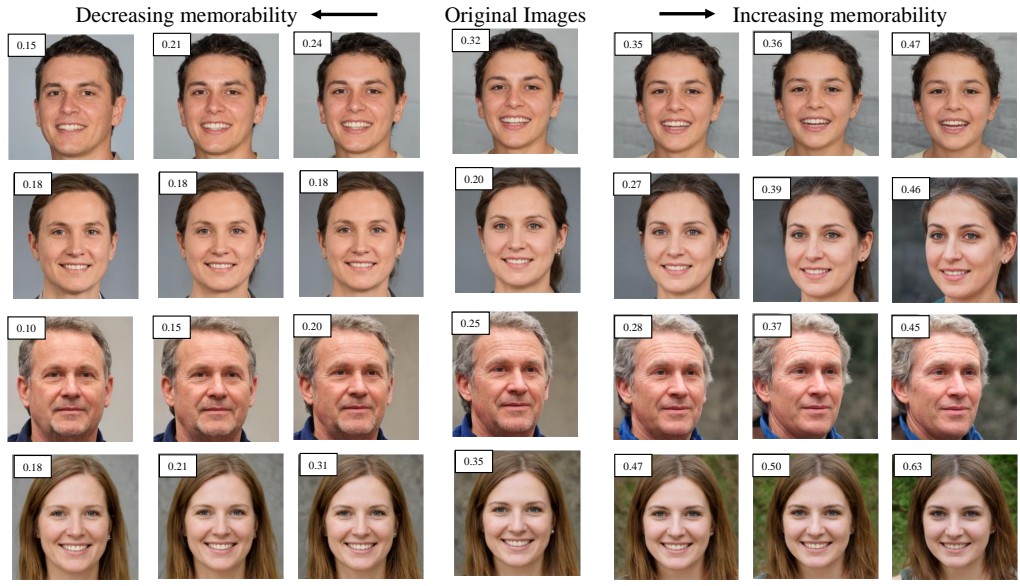

Figure 15: **Generated images by StyleGAN2 with their corresponding memorability scores.** This figure shows the modified images when latent space of the StyleGAN2 was used to determine the separating hyperplane. Square-shaped faces were fed to the assessor.

## A.2 LAYERWISE MODIFICATIONS

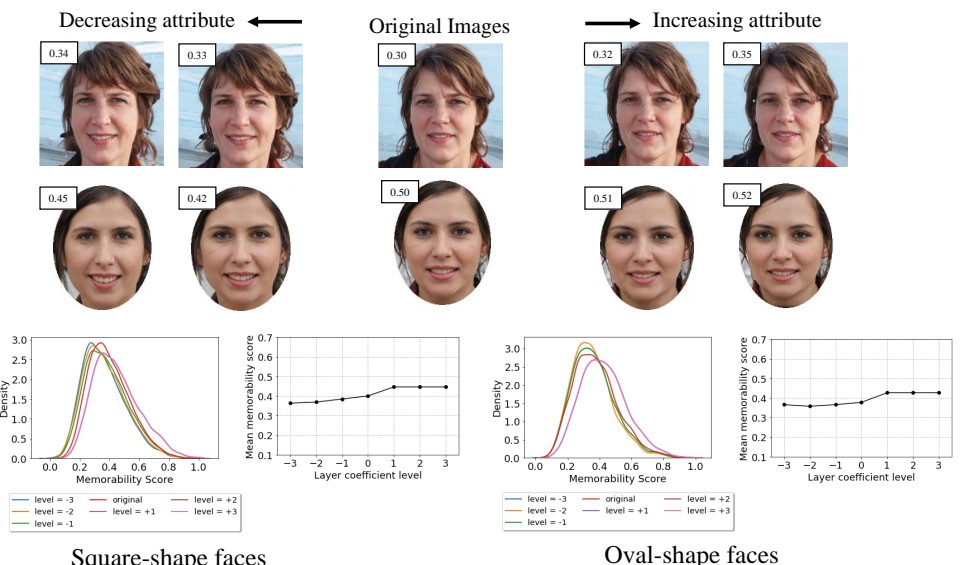

Figure 16: **The effect of the changes in the 1st layer of the extended latent space on memorability score.** This layer mostly affects shape of the face. Increasing this attribute makes the face smaller, whereas decreasing this attribute makes the face larger, which usually may cause a decrease in the memorability score.

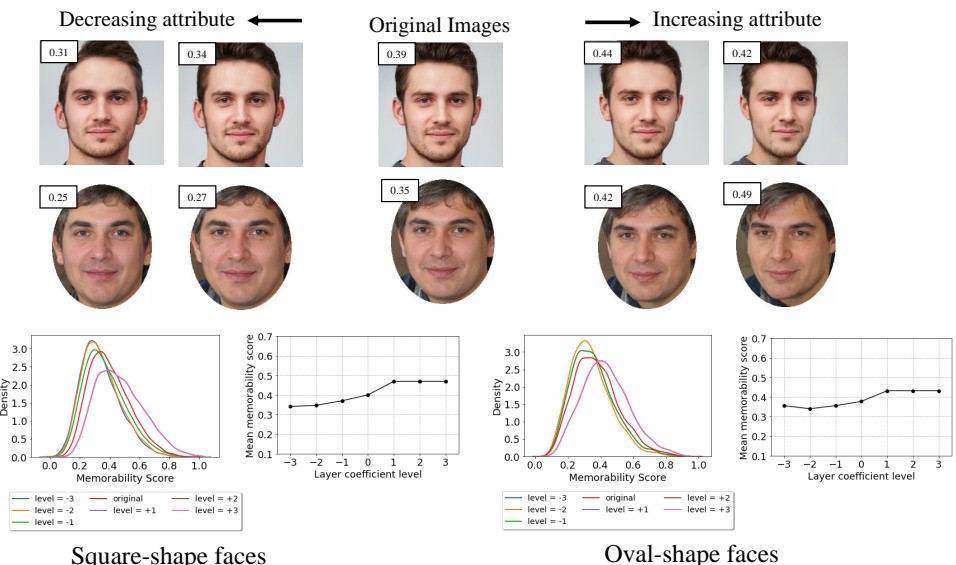

Figure 17: **The effect of the changes in the 2nd layer of the extended latent space on memorability score.** This layer mostly affects hair, pose of the face, and the direction of the eyes and shows how these attributes contribute to the memorability score.

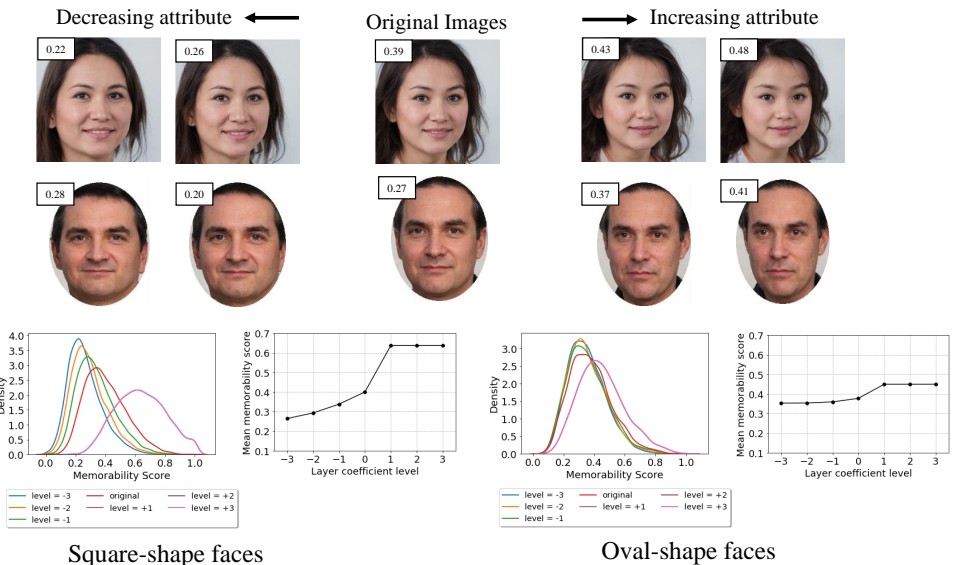

Figure 18: **The effect of the changes in the 3rd layer of the extended latent space on memorability score.** This layer mostly affects shape and seriousness of the faces. We observed that moving this layer, in the positive direction of the corresponding layer in the memorability vector ($w^*$), will highly increase the memorability score.

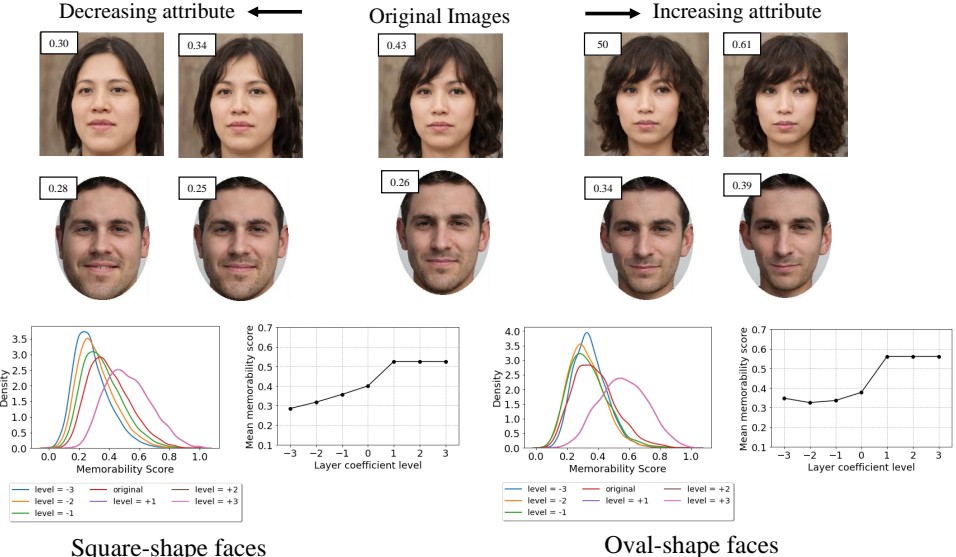

Figure 19: **The effect of the changes in the 4th layer of the extended latent space on memorability score.** This layer mostly affects shape of the face (especially the chin). We can observe that changes in this attribute, largely contribute to the memorability of the face and moving this layer in the positive direction of the corresponding layer in the memorability vector ($w^*$), will increase the memorability score.

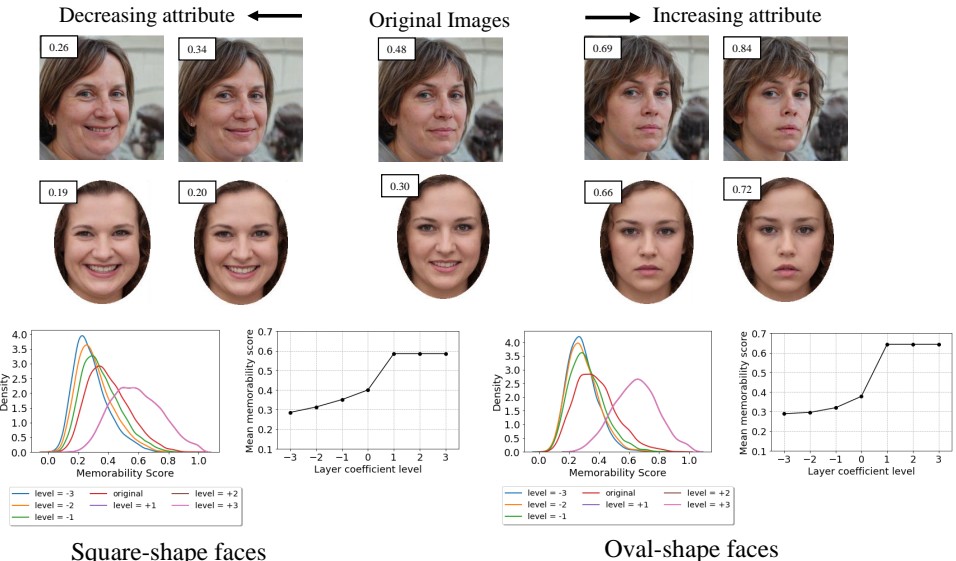

Figure 20: **The effect of the changes in the 5th layer of the extended latent space on memorability score.** This layer mostly affects nose, lips, facial weight, and smile. This is one of the most important layer that plays a role in determining the memorability score.

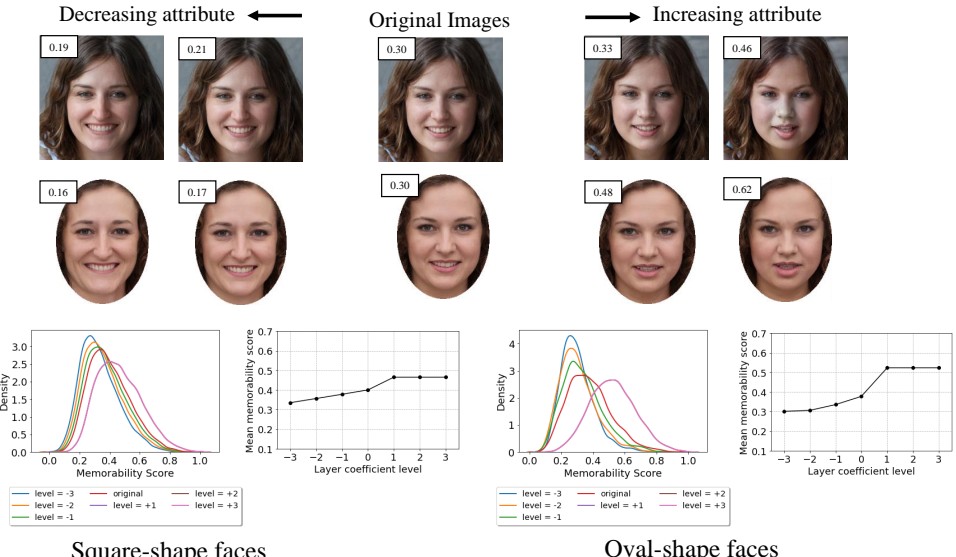

Figure 21: **The effect of the changes in the 6th layer of the extended latent space on memorability score.** This layer mostly affects the smile and form of the lips. We observe that when we move this layer, in the positive direction of the corresponding layer of the memorability vector $(w^*)$, the memorability score will increase hugely and the person's lips will become thicker. However, when you move it in the opposite direction, the person's lips will become thinner and the memorability score will decrease.

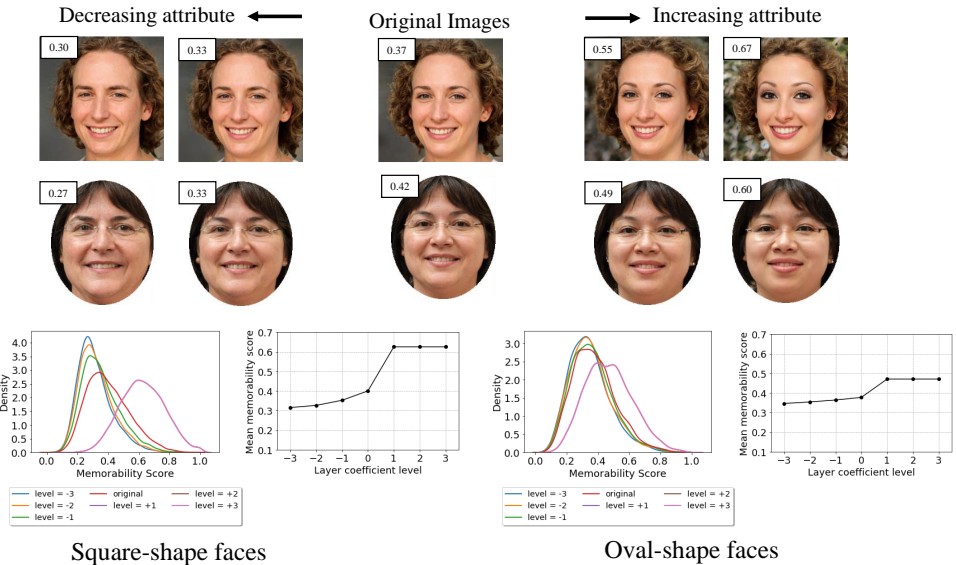

Figure 22: **The effect of the changes in the 7th layer of the extended latent space on memorability score.** This layer mostly affects make-ups and facial hair. We observe that when we move this layer, in the positive direction of the corresponding layer of the memorability vector ($w^*$), the memorability score will drastically increase and the makeup starts to appear on the person's face. The lips will become thicker and the eyebrows shape and the person's skin changes.

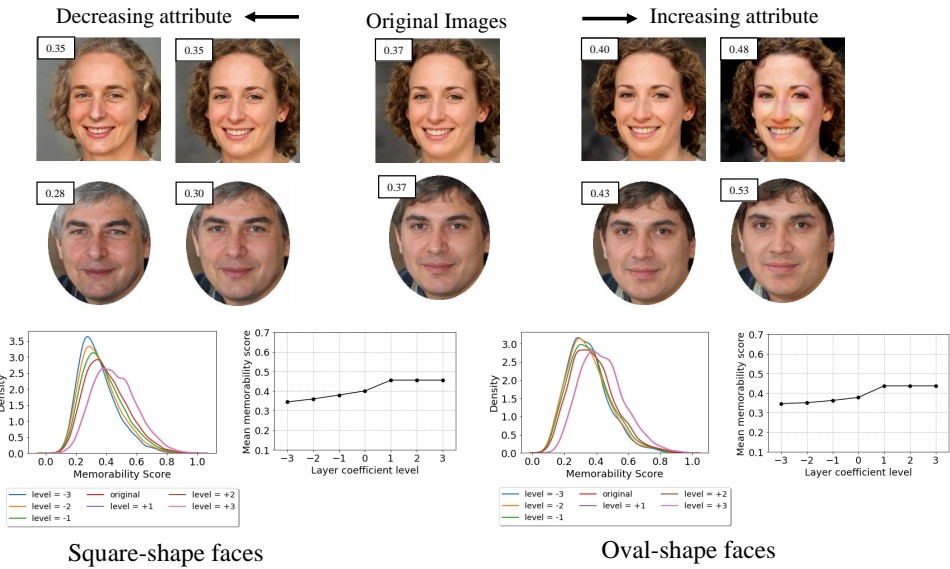

Figure 23: **The effect of the changes in the 8th layer of the extended latent space on memorability score.** This layer mostly affects the eyes and the hair color.

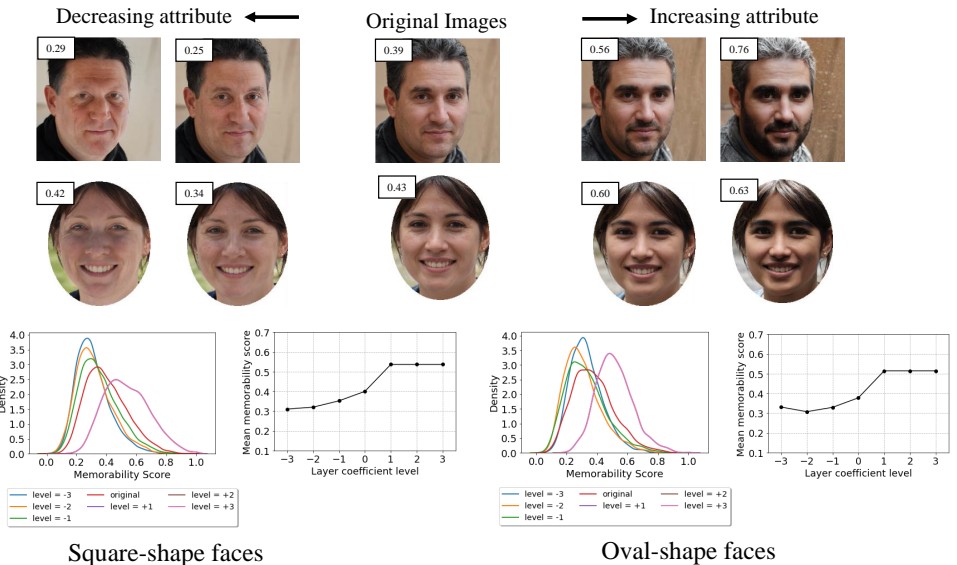

Figure 24: **The effect of the changes in the 9th layer of the extended latent space on memorability score.**This layer mostly affects the facial hair, hair color type and skin color. moving this layer, in the positive direction of the corresponding layer of the memorability vector ($w^*$), will make the hair color gold and some shadows and facial hair (if the person is male), will appear on the face.

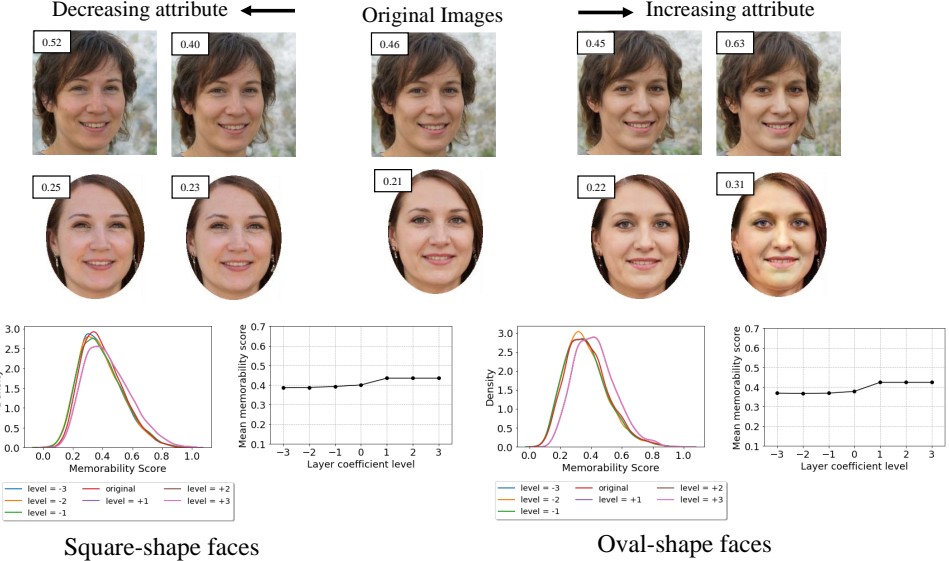

Figure 25: **The effect of the changes in the 10th layer of the extended latent space on memorability score.** The changes in this layer or mostly responsible for modifications on skin color and eyes. We observe, the changes in this layer are not as effective as other layers to modify the memorability score.

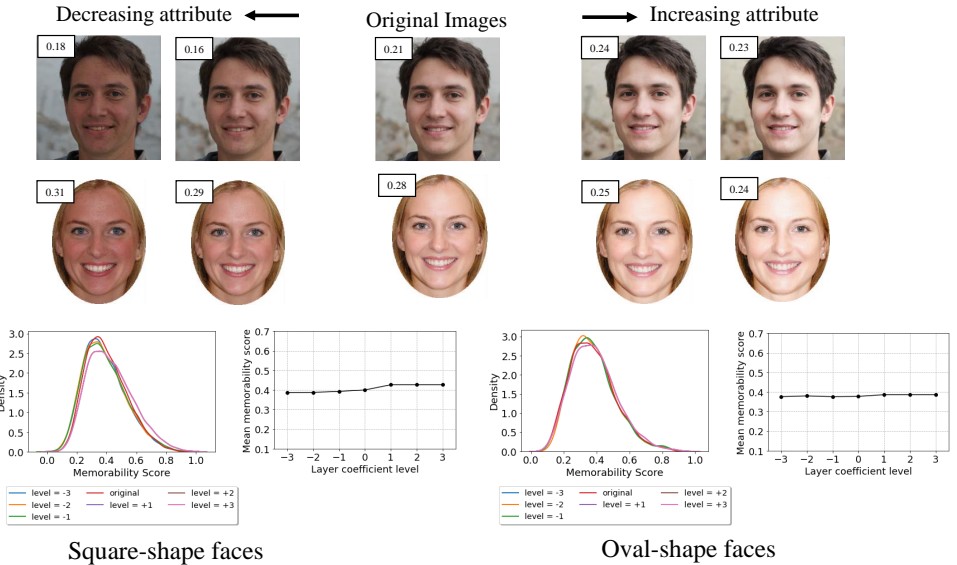

Figure 26: **The effect of the changes in the 11th layer of the extended latent space on memorability score.** Changes in this layer control the skin color and face brightness. As it is shown, this layer does not play an important role in modifying memorability score of the face.

### A.3    FACE MEMORABILITY ASSESSORS

In order to train the models, we split the 10k US Face Database images into train, validation and test split. We used 80 percent of the data as the training samples and used 10 percent of the data for each of the test and validation splits. Euclidean distance was used as the loss function and the batch size was set to 64. Moreover, we leveraged Adam optimizer to train our models. Due to large false alarm rates in human face images, we trained our models both with raw memorability scores (computed by hit rate) and corrected memorability scores (considering false alarm rate). We also tried some simple augmentations on the dataset and found, the score of the models will slightly increase if we use a simple augmentation like random horizontal flipping($p = 0.5$).

Consistent with Khosla et al. Khosla et al. (2013b), we observed when the corrected hit rate scores are used, all models outperform the case when only hit rate scores are used. That is because false alarm rate introduces noise to memorability scores, therefore, the models perform better when we reduce the noise by correcting for false alarms. We have brought the rank correlation scores using pretrained VGG16, ResNet50 and SENet50 using hit rate and true hit rate values.

Table 3: Memorability scores of the models pre-trained on face recognition (on VGGFaces database) and fine-tuned on 10K US face database. Note that all these computational models, produce larger Spearman's rank correlation score when true hit rate scores are used.

| Model | Hit Rate Score | True Hit Rate Score |
|---|---|---|
| VGG16 | 0.445 | 0.579 |
| ResNet50 | 0.433 | 0.607 |
| SENet50 | 0.448 | 0.601 |

### A.4    FINDING HYPERPLANE IN THE LATENT SPACE

In this work, we have provided the results of the separating hyperplane accuracy in extended latent space. We utilized the latent space to find the separating hypeprlane in the latent space of StyleGAN2 and reported the accuracy of the hyperplane.

Table 4: Accuracy of the separating hyperplane, based on the method for dividing images into high-memorable and low memorable images, the shape of the images, and the assessor. In this case latent space ($\mathbb{R}^{512}$) is used to find the separating hyperplane. We can observe that the accuracy of the hyperplane is lower than the case when extended latent space is used.

| Assessor | Median | | Mean | |
|---|---|---|---|---|
| | Oval | Square | Oval | Square |
| ResNet50 | 0.699 | 0.686 | **0.706** | 0.683 |
| SENet50 | 0.696 | 0.723 | 0.700 | **0.733** |
| VGG16 | 0.689 | 0.695 | 0.697 | 0.705 |

## A.5 MODIFYING THE FACES CONDITIONALLY

As we described previously, one of the benefits of our work is that it makes it possible to modify the memorability scores of the faces conditionally. Different hyperplanes for different face attributes could be used and then with projection, we can try to maintain the corresponding attributes unchanged. As there is a correlation between different face attributes, choosing large weights and attempting to change the memorability scores drastically, may affect the attribute. We showed the attempt to maintain smile and age attributes in Figure 27 as an example. However, this method could be applied to a variety of face attributes.

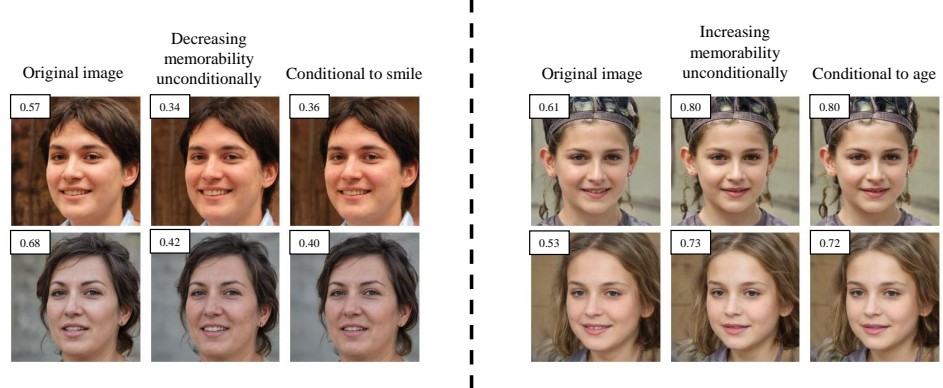

Figure 27: **Modifying memorability scores of the faces, with the condition to maintaining smile and age.** This figure shows how projection can affect memorability and the special attribute that we are aiming to maintain unchanged.

## A.6 MODIFYING MEMORABILITY OF GENERATED NON-FACE IMAGES BY STYLEGAN

We extended our experiments and tested our method on other non-face images. We leveraged pretrained StyleGAN2 on churches, cats, horses and cars. In here, we used MemNet as our memorability assessor. (See Table 5 for accuracy of the hyperplanes in churches, cats, horses and cars augmented latent space.)

Table 5: Accuracy of the separating hyperplane, based on the weights of the generator.

| Weight | Accuracy |
|---|---|
| Cats | 0.7875 |
| Horses | 0.8897 |
| Cars | 0.8581 |
| Churches | 0.8325 |

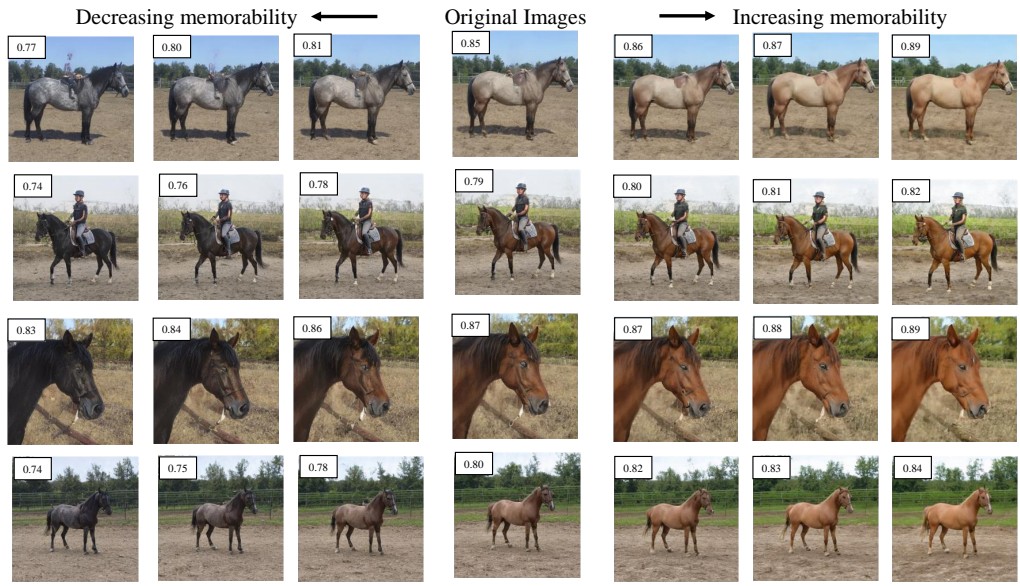

Figure 28: **Generated horse images by StyleGAN2 with their corresponding memorability scores.** This figure shows the modified images when extended latent space of the StyleGAN2 was used to determine the separating hyperplane.

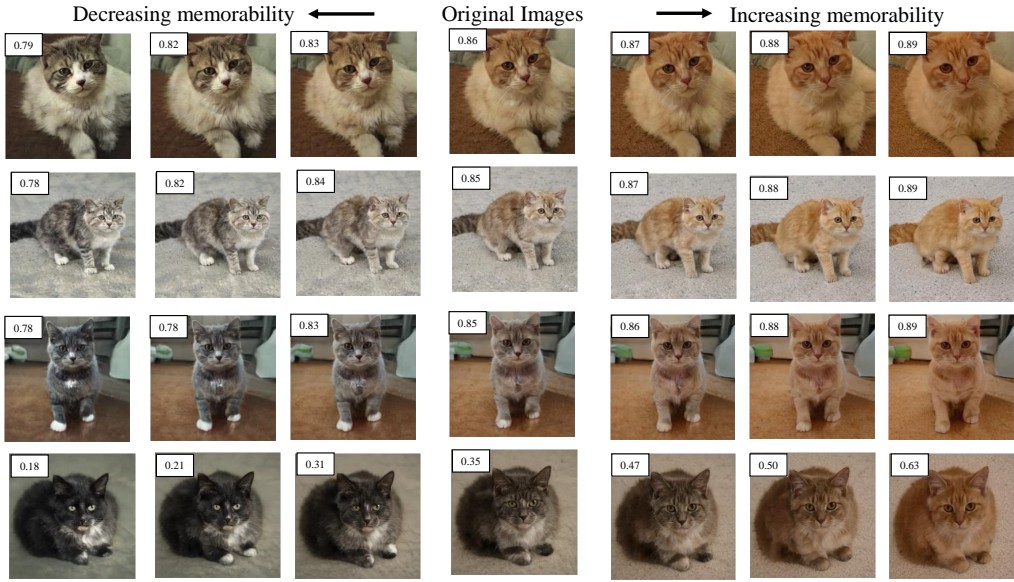

Figure 29: **Generated cat images by StyleGAN2 with their corresponding memorability scores.** This figure shows the modified images when extended latent space of the StyleGAN2 was used to determine the separating hyperplane.

## A.7 MODIFYING MEMORABILITY OF GENERATED OBJECT IMAGES BY BIGGAN

Lastly, we tried to show the effectiveness of our method on generated images by BigGAN. We generated 200k images by $512 \times 512$ BigGAN-deep (Brock et al., 2018), predicted their memorability scores by our assessor, and divided them into low-memorable and highly-memorable images. We observed that the effect of the modifications. is similar to Goetschalckx et al. (2019). Increasing the

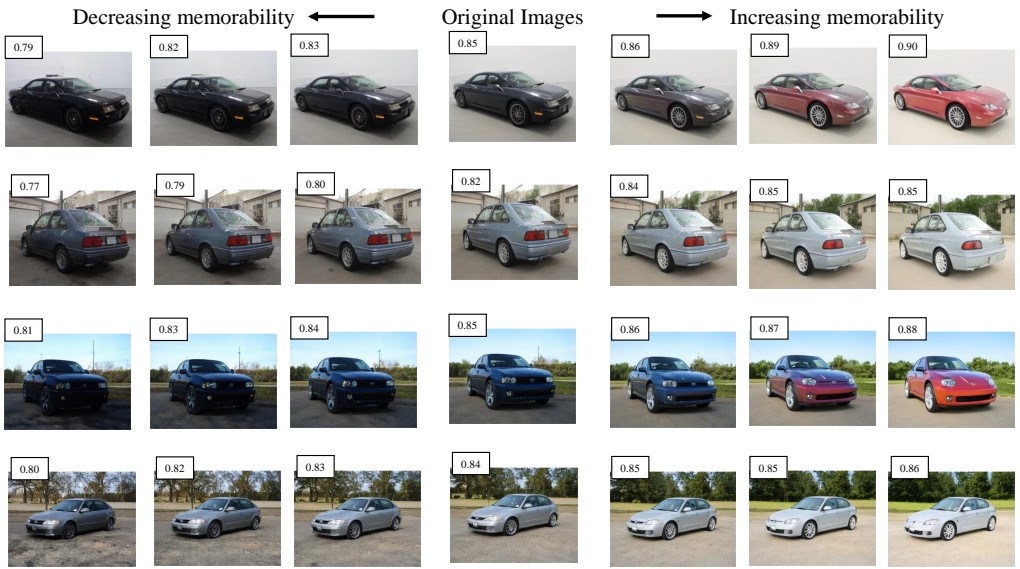

Figure 30: **Generated car images by StyleGAN2 with their corresponding memorability scores.** This figure shows the modified images when extended latent space of the StyleGAN2 was used to determine the separating hyperplane.

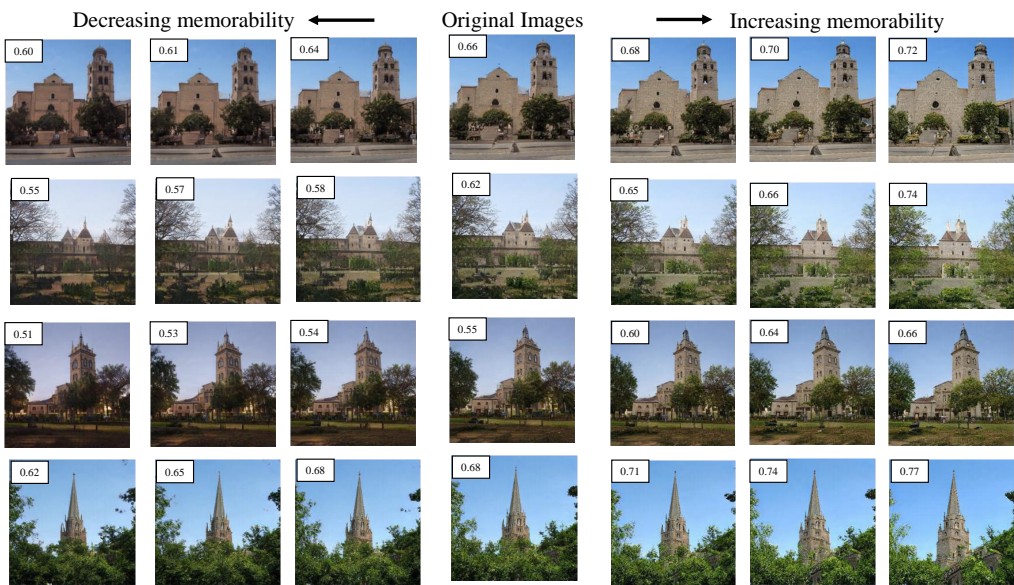

Figure 31: **Generated church images by StyleGAN2 with their corresponding memorability scores.** This figure shows the modified images when extended latent space of the StyleGAN2 was used to determine the separating hyperplane.

memorability scores, caused the images to become zoomed-in, in some cases the color changed and also in a few cases (Cheese burger and snake in Figure 35a, made the objects rounder.

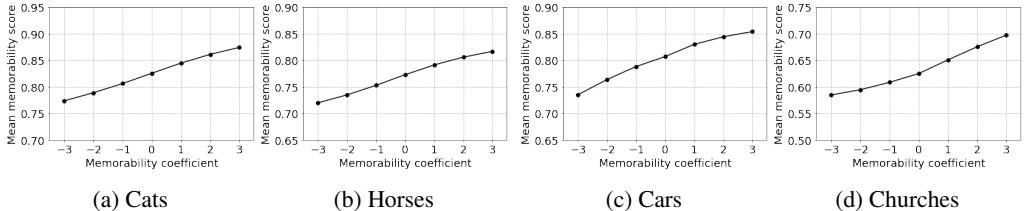

(a) Cats      (b) Horses      (c) Cars      (d) Churches

Figure 32: The effectiveness of our method for modifying memorability scores tested on 5k generated images for each category. As depicted the distribution and mean memorability score of images changes with the coefficient used for memorability modification

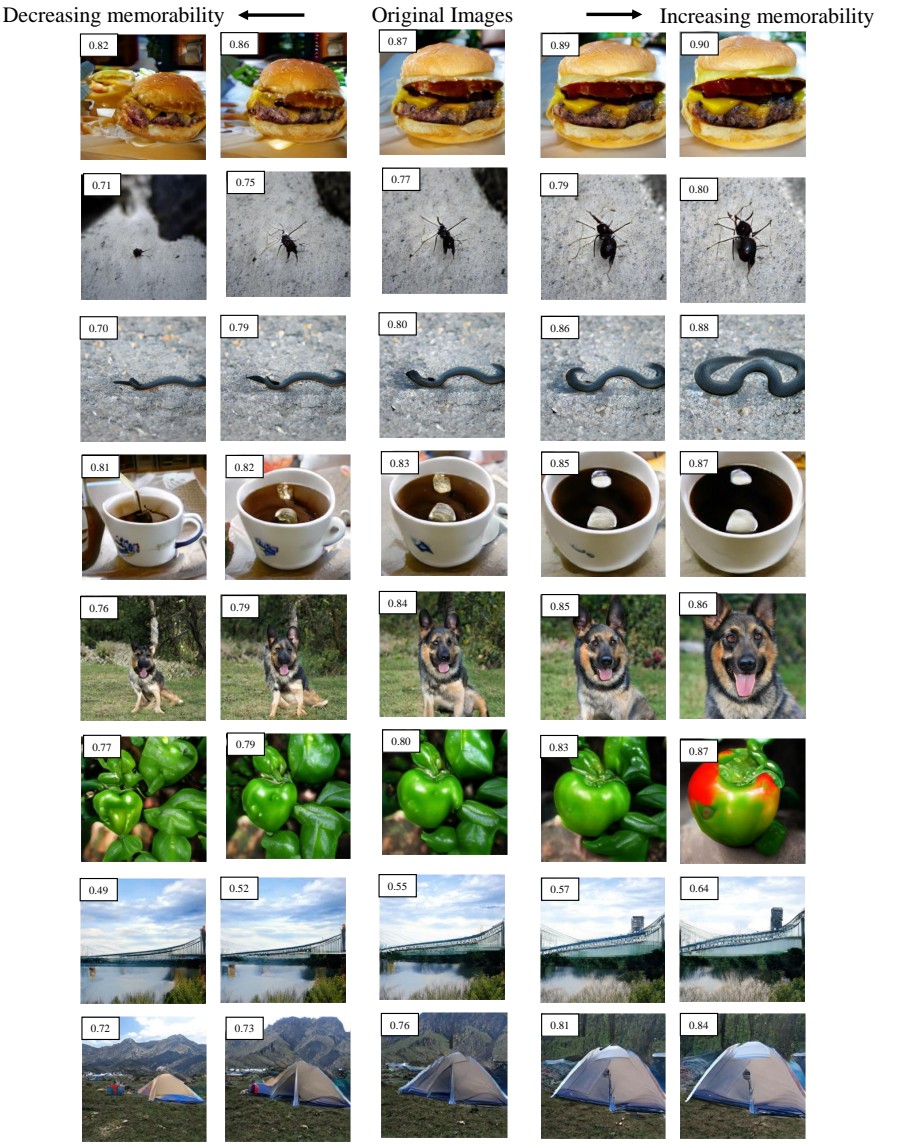

Figure 33: **Generated images by** $512 \times 512$ **BigGAN-deep with their corresponding memorability scores.** This figure shows the modified images when the latent space of the BigGAN ($\mathbb{R}^{128}$) was used to determine the separating hyperplane.

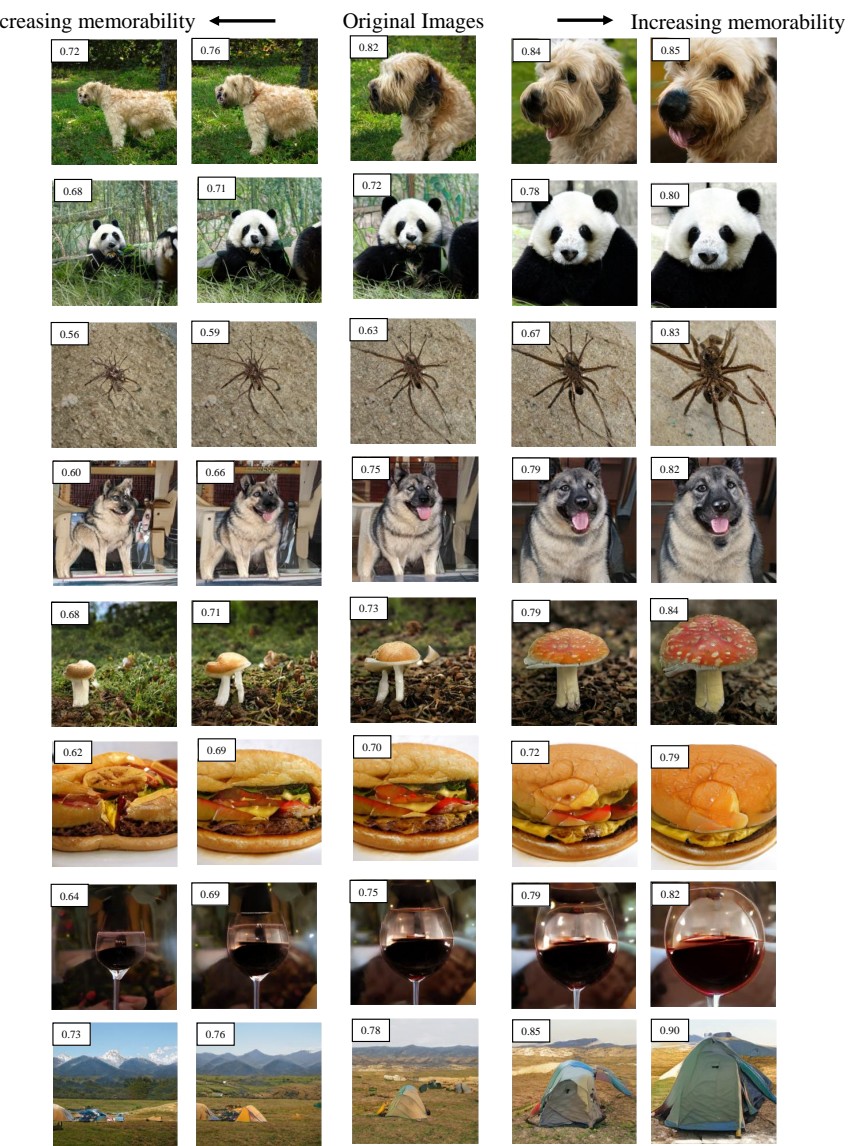

Figure 34: **Generated images by** $256 \times 256$ **BigGAN with their corresponding memorability scores.** This figure shows the modified images when the latent space of the BigGAN ($\mathbb{R}^{140}$) was used to determine the separating hyperplane.

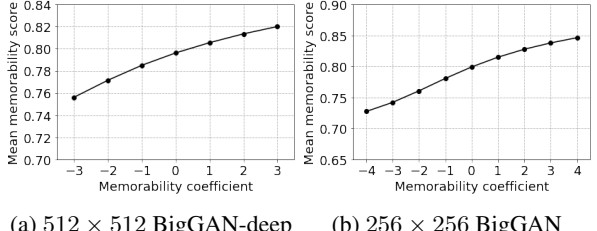

(a) $512 \times 512$ BigGAN-deep  (b) $256 \times 256$ BigGAN

Figure 35: The effectiveness of our method for modifying memorability scores tested on 20k generated images. As depicted the distribution and mean memorability score of images changes with the coefficient used for memorability modification

