# OpenReview forum: "CONTROLLING THE MEMORABILITY OF REAL AND UNREAL FACE IMAGES"
_ICLR.cc/2022/Conference — ICLR 2022 Submitted_

### Official Review · Reviewer_8EvC · 2021-10-26

**Correctness:** 3
**Technical Novelty And Significance:** 1
**Empirical Novelty And Significance:** 2
**Recommendation:** 3
**Confidence:** 4

**Main Review:**

## Strengths

- The proposed approach is reasonably designed to modify face memorability.

- The paper shows many example images, and we can observe some interesting cases of appearance changes with respect to memorability scores.

## Weaknesses

- Technical novelty is not clear. As discussed in the paper, the underlying approach based on the attribute-separating hyperplane is mostly equivalent to InterFaceGAN. Shen et al. already demonstrated their approach on face image generation tasks, and this work extends it to memorability modification. The concept of image memorability modification for generated faces has been also discussed in [Goetschalckx et al., 2019], and the contribution is not significant from the technical perspective.

- Experimental validation is not sufficient. Essentially, this paper provides qualitative results and quantitative evaluation in terms of face realness (FID and KID metrics) and memorability (assessor score) without any comparison with baseline approaches. There are some missing aspects such as memorability assessment with human participants and comparison with existing memorability modification approaches. In both senses, it would be required to demonstrate the difference from [Goetschalckx et al., 2019] by showing that the proposed method can control human memory performance on face images better.

- While the paper claims that the proposed technique aims at modifying memorability without affecting identity and other facial attributes, there are some cases in the examples whose identity clearly changed after memorability modification. Compared to other relatively objective facial attributes, it is not a trivial task to judge if the facial identity was maintained during the modification and this would require quantitative evaluation by either machine or human evaluator.

**Summary Of The Paper:**

This paper presents a method to control the memorability of face images in the latent subspace of GANs. The basic approach is similar to InterFaceGAN [Shen et al., 2020]. The authors first train a memorability assessor network on the 10k US Adult Faces Database and used the assessor to provide memorability scores on generated face images from StyleGANs pre-trained on the FFHQ dataset. They then find the hyperplane in the StyleGAN latent space to separate low-memorable and high-memorable images according to the memorability score. The memorability of faces is modified by changing the distance from the hyperplane. They demonstrated that their method can successfully control the memorability scores of generated images while keeping the realism as face images and show some qualitative examples and discussions about memorability-modified face images.

**Summary Of The Review:**

The approach is reasonable and has the potential to lead to new findings in face memorability studies. However, given that the technical novelty is relatively incremental and the experimental results obtained at this point are still in the early stages, the research contribution is not sufficient.

---

> ### Author Response · Authors · 2021-11-22
> **Respond to Reviewer t o8EvC**
>
> Thank you for your extensive comments.
>
> * To the best of our knowledge, no study exists on modifying the memorability of real faces. Compared to [Goetschalckx et al., 2019], [Sidorov., 2019], our method is much faster and more efficient in terms of memory and storage. To summarize the novelties in our research,
>     + Ours is the first work that attempted to modify the memorability score of real faces and we were successful in doing so.
> We trained three new models for predicting the memorability score of the face images (our assessors). To the best of our knowledge, there is no publicly available model for predicting face memorability.
>     + Although we were inspired by InterFaceGAN [Shen et al., 2020], there are some major differences. Firstly, InterFaceGAN is used for some binary attributes (e.g., age, gender, and wearing or not wearing eyeglasses) for which there is a binary classifier. However, in our work, we used the memorability scores of faces (which are continuous) and analyzed their distribution to divide them into two groups. In addition, **it is not trivial that there exists a separating hyperplane based on the memorability scores**, as the memorability score is a combination of the different face attributes.
>     + Previous work (Isola et al. 2013; Khosla et al. 2015) showed the rank of the memorability scores for the images is highly consistent across participants, and time intervals. In other words, they showed that while the memorability values may change across time delays, the rank of memorability scores is highly preserved. Therefore, in the field of memorability, the rank of memorability scores are treated as time-independent, and observer-independent measures. As such, following previous works, we opted to use the scores from the images themselves to separate them and focused on the rank of the memorability scores. Hence, we can say that our method is robust to different assessors. However, Ganalyze uses the exact values of memorability scores which are not time and observer-independent and may cause some problems. Consider a case when there are two categories of images. When the range of memorability scores is low across one category and high in another, it will greatly affect the realness of the image if using the Ganaylze loss ($E[(A(G(T(z,a))) - (A(G(z))-a))^2]$). Moreover, as the exact value of memorability is used, it cannot be used for square-shaped faces, because all the faces in the 10k us face database are oval-shaped and the assessor will be trained on oval-shaped faces and cannot predict the exact memorability scores of square-shaped faces very well. Although, it can predict the rank of the images.
>     + By proposing our hyperplane method, we are able to modify the memorability score of the faces conditionally. Using the projection method explained in the paper, we can conditionally modify the memorability score while fixing selected attributes such as pose, smile, glasses, age, etc.
> * To measure the realness of our modified faces, we compared them against the unchanged faces. We compared the FID and KID scores of the modified images with unmodified images as our baseline. In creating this comparison, we use StyleGAN2 because it is the state-of-the-art model for generating realistic faces. While we acknowledge that a  human memory experiment will be beneficial. Instead, we train memorability models for predicting face memorability scores and test our method on them. Running human experiments is expensive and time-consuming and there is a whole body of previous studies that have shown that memorability of images can be predicted by deep neural networks to the same level of human performance. In this work, we find a direction in the latent space such that moving in said direction will change the memorability score. The weight for moving in that direction is controlled by the user. Furthermore, there is a range for changing the memorability score such that the identity is not changed. In our extensive experiments, we observed that if you move the latent space with a small weight, the identity will not be affected.

---

> > ### Comment · Reviewer_8EvC · 2021-11-26
> > **Response to Authors**
> >
> > Thank you for the detailed response.
> >
> > - I understand the technical difference from prior work, but in my opinion, these claims at least require experimental support with human subjects. Although I agree with the authors that the evaluation using memorability models shows validity to some extent, this sounds a little tautological to the reviewer because the hyperplane IS found with respect to the memorability scores predicted by the same model. We cannot completely rule out the possibility that this is a shortcut to the memorability model.
> > - It was not clear if the application of InterFaceGAN to a continuous attribute is a technical novelty. If it is not trivial that a separating hyperplane exists here, this sounds even more like it needs to be evaluated by humans.
> > - The same discussion will apply to the face identity. To claim that the method can modify memorability without affecting the identity, the paper should have a quantitative evaluation (ideally with a human evaluator). It will be an important question how much we can control memorability (with "small weights") while maintaining the identity.
> > - As also pointed out by reviewer iYVp, quantitative comparison with existing methods is another critical missing aspect.

---

> > > ### Author Response · Authors · 2021-11-29
> > > **Response to Reviewer 8EvC**
> > >
> > > Thank you for your comment.
> > >
> > > The memorability models have been tested on the face database and we have brought their results in A.3. The results show that these models are promising for predicting the memorability score of the faces. Moreover, the qualitative results are similar to [Sidorov., 2019], which suggests that we are successful in modifying the memorability scores.
> > >
> > > In this work, our **focus** is on the **analysis and modification of face image memorability**. There is no prior work that we can compare our work with it. [Siarohin et al., 2017] uses different styles and adds those styles to images, the manipulated images do not look real. [Sidorov., 2019] trains an encoder and can only modify the memorability scores in three discrete levels. However, in our work, we leverage a weight parameter to modify the memorability score of the face continuously. Moreover, [Goetschalckx et al., 2019] trains a transformer to modify the memorability scores of only object images generated by BigGAN. At a higher level, our method and [Goetschalckx et al., 2019] are following similar principles, as we both move the latent vector in a direction to modify the memorability score. However, we showed that there exists a separating memorability hyperplane in the latent space of the GANs that can be used to modify the memorability scores. In contrast with [Goetschalckx et al., 2019], in our method, less data is required and it is much faster and also suggests that we can use conditional memorability modification (last paragraph of 2.4). In addition, comparing it with other methods is not well defined as in our work there is a memorability modification weight that we can leverage to modify the memorability to an arbitrary value

---

### Official Review · Reviewer_xi9k · 2021-10-28

**Correctness:** 3
**Technical Novelty And Significance:** 2
**Empirical Novelty And Significance:** 2
**Recommendation:** 6
**Confidence:** 4

**Main Review:**

As pros:
 - basic observation (latent space analysis with effect on memorability) is very interesting. it is strong enough to make the paper outstanding. However the paper develops ideas previously published as mentioned in prior work
- technical bases is solidly built
- the evaluation is rather strong. The observation, which again is very interesting, needs to be proven by experimental evaluation and the paper does a good job. Furthermore, all the examples  showed in the paper do confirm the basic idea and developments
- the appendix is useful

As cons:
 - the paper is limited in scope as it addresses only the problem of face memorability
- the experimentation is limited to the statistic of UsFaces database. It is not extended with an analysis over gender, age, race.
- the evaluation does not compare the proposed method with prior art mentioned on pager 3 in paragraph "Modifying Image Memorability."

Small issues:
 - the amount of images used in experimentation is "hidden" in the text. The text should me be made more clear to enhance the quantities used. Also some of the quantities should be added in the figure and tables captions

**Summary Of The Paper:**

The paper addresses the problem face image memorability. The proposal is based upon the identification in Style GAN of a hyper axis in the latent space where movement is correlated with memorability

**Summary Of The Review:**

Overall the idea of of the paper is nice but the evaluation is barely convincing.

---

> ### Author Response · Authors · 2021-11-22
> **Respond to Reviewer xi9k**
>
> Thank you for your comments and interest in our paper.
>
> * In this work, our focus is on the analysis and modification of **face** image memorability. We use StyleGANs in our work because they are state-of-the-art models for generating **real-looking faces**. Our utilization of StyleGANs is required to create a dataset of realistic-looking faces. We use this generated dataset to find the memorability hyperplane in the model’s latent space. Not only that, for modifying the memorability of real faces, we need StyleGANs to **reconstruct real faces with high accuracy**. Moreover, we want to analyze the relationship between memorability scores and facial attributes. We aim to see what attributes of a face contribute to the memorability score. We also analyze what facial attributes change when we modify the memorability score. Next, StyleGANs provide an extended latent space which we leverage to derive a more accurate memorability hyperplane. Also, the face attributes of StyleGANs are especially disentangled in comparison to other GANs, which is required to accurately modify faces for memorability and study attributes that contribute to this modification. These are the reasons why we chose StyleGANs as the generator in our work. This is now clarified in the revised manuscript.
> * In expanding our method and showing its validity and effectiveness, we added new experiments on the memorability of non-face objects in the appendix. We use StyleGAN2 independently pre-trained on cars, churches, horses, and cats to generate images in these categories. With our method, we are still able to modify the memorability scores of such objects. To show the effectiveness of our method, we mimic our previous experiments by generating 5k images in each category, modifying their memorability scores with different weights, and plotting the mean of their memorability scores.
> * To show that our method does not rely on the StyleGAN architecture, we add new experiments in the appendix where we apply our method on object images generated by BigGANs. We demonstrate the effectiveness of our method by testing it with different weights on 20k generated images by BigGANs.
> * While we acknowledge that analysis of race and gender are important in the analysis of faces, but in our work, we are limited to the only memorability face dataset available to date, the 10K US face dataset. Future works are indeed required to develop larger face memorability datasets where further related experiments and analysis are possible.
> We talked about their limitations in the revised paper and added another experiment on generated objects by BigGAN.
> Thanks for reading our paper carefully. We have revised our paper to make it more clear and added more quantitative information to Figure and Table captions.

---

> > ### Comment · Reviewer_xi9k · 2021-11-25
> > **Keep the recommendation**
> >
> > Thank you for your careful response to my questions.
> >
> > I see the point of having insufficient available data (limited to only 10K US face dataset) yet this is an external limitation  and is less convincing about the the strength of the current paper. I still believe that the paper is interesting but only marginally.

---

> > > ### Author Response · Authors · 2021-11-29
> > > **Response to Reviewer xi9k**
> > >
> > > Thank you for the comment.
> > >
> > > As you mentioned, it is a limitation of the dataset that we cannot analyze the memorability scores based on race. Despite that, we have added new experiments to the appendix to show the power of our method and addressed your previous concerns about the scope of the paper.

---

### Official Review · Reviewer_iYVp · 2021-11-02

**Correctness:** 2
**Technical Novelty And Significance:** 2
**Empirical Novelty And Significance:** 2
**Recommendation:** 3
**Confidence:** 4

**Main Review:**

Strengths:
1. The experimental results show that the proposed method does change the memorability of images.
2. The proposed method can be applied to real facial images.

Weaknesses:
1. Are there experimental comparisons with other existing methods that change image memorability? Can those methods be applied to the images generated by StyleGAN? What're the advantages of the proposed method over the existing methods?
2. When choosing VGG16, ResNet50, or SENet50, why the accuracy of them on the US 10k Face Database with real labels is not considered?
3. The accuracy when using the extended latent space (w) is reported in Table 2. It's better to report the accuracy when using the latent space (z) in the appendix.
4. Fig 4, 5, 6, and 8 lack analysis, such as what can be proved by these figures.  The top-left numbers on the facial images are not explained.

**Summary Of The Paper:**

This paper proposes to find the hyperplane to separate images with different memorability in the latent-vector spaces of StyleGAN1 and StyleGAN2, then they can move the latent-vectors along the normal vector of the hyperplane to control the memorability of the generated images. A SENet50 trained on the US 10k Face Database with memorability labels is used to find the hyperplane. The experimental results show that the proposed method can change the memorability of images measured by the scores of the SENet50, and this method can be extended to modify real facial images.

**Summary Of The Review:**

The effectiveness of the proposed method is validated by experiments, while the superiority (over existing methods) is not. The experimental results need more detailed descriptions.

---

> ### Author Response · Authors · 2021-11-22
> **Respond to Reviewer iYVp**
>
> Thank you for your thoughtful and accurate comments.
>
> * We added some new experiments in the appendix and additionally applied our method on object images generated by BigGANs. We demonstrated the effectiveness of our method by testing it with different weights on 20k generated images by BigGANs. You can observe that the results are similar to the ganalyze[Goetschalckx et al., 2019]. However, our method is much faster and more efficient in terms of memory and storage. On the other hand, Previous work (Isola et al. 2013; Khosla et al. 2015) showed the rank of the memorability scores for the images is highly consistent across participants, and time intervals. In other words, they showed that while the memorability values may change across time delays, the rank of memorability scores is highly preserved. Therefore, in the field of memorability, the rank of memorability scores are treated as time-independent, and observer-independent measures. As such, following previous works, we opted to use the scores from the images themselves to separate them and focused on the rank of the memorability scores. Hence, we can say that our method is robust to different assessors. However, Ganalyze uses the exact values of memorability scores which are not time and observer-independent and may cause some problems. Consider a case when there are two categories of images. When the range of memorability scores is low across one category and high in another, it will greatly affect the realness of the image if using the Ganaylze loss ($E[(A(G(T(z,a))) - (A(G(z))-a))^2]$). Moreover, as the exact value of memorability is used, it cannot be used for square-shaped faces, because all the faces in the 10k us face database are oval-shaped and the assessor will be trained on oval-shaped faces and cannot predict the exact memorability scores of square-shaped faces very well. Although, it can predict the rank of the images.
>
> * In evaluating memorability models, we always calculate Spearman’s rank correlation between the true memorability scores and the predicted memorability scores. Because based on previous studies in the field of image memorability,  the rank score is consistent across different time delays and participants, the rank and relation between memorability scores are more important than the memorability values themselves. We added the rank score of our face memorability models (SENet, ResNet, and VGG16) which is a measure of their accuracy scores to the appendix.
>
> * We do agree with this comment and added the table reporting accuracy in z space to the appendix.
>
> * Thank you for your comment. We revised our manuscript and added more explanations about the figures.

---

> > ### Comment · Reviewer_iYVp · 2021-11-25
> > **Response to Author**
> >
> > Thank you for your careful response to my questions.
> > 1. Although the experimental performance of the proposed method on BigGAN is reported, the comparison to existing methods is still absent. The key point is that, because "change image memorability" is not an unprecedented task, the proposed method has to be compared to existing methods to prove that it is worth noticing while we already have those existing methods. The advantages of the proposed method over existing methods need to be shown in experiments rather than general descriptions.

---

> > > ### Author Response · Authors · 2021-11-29
> > > **Response to Reviewer iYVP**
> > >
> > > Thank you for your comment.
> > > In this work, our **focus** is on the **analysis and modification of face image memorability**. There is no prior work that we can compare our work with it. [Siarohin et al., 2017] uses different styles and adds those styles to images, the manipulated images do not look real. [Sidorov., 2019] trains an encoder and can only modify the memorability scores in three discrete levels. However, in our work, we leverage a weight parameter to modify the memorability score of the face continuously. Moreover, [Goetschalckx et al., 2019] trains a transformer to modify the memorability scores of only object images generated by BigGAN. At a higher level, our method and [Goetschalckx et al., 2019] are following similar principles, as we both move the latent vector in a direction to modify the memorability score. However, we showed that there exists a separating memorability hyperplane in the latent space of the GANs that can be used to modify the memorability scores. In contrast with [Goetschalckx et al., 2019], in our method, less data is required and it is much faster and also suggests that we can use conditional memorability modification (last paragraph of 2.4). In addition, comparing it with other methods is not well defined as in our work there is a memorability modification weight that we can leverage to modify the memorability to an arbitrary value.

---

### Official Review · Reviewer_m2GG · 2021-11-02

**Correctness:** 3
**Technical Novelty And Significance:** 2
**Empirical Novelty And Significance:** 3
**Recommendation:** 5
**Confidence:** 3

**Details Of Ethics Concerns:**

No ethincal issue

**Main Review:**

Strength

+ They tackle an interesting problem “What makes an image memorable?”, where the problem is well defined. This work is the first experiment about ‘memorability modification’ that could be applied in various applications.

+ Their approach solves the problem by introducing the concept of distance between the latent vector or the extended latent vector and the normal vector of the hyperplane.

+ The paper is easy to follow and presented the experimental results with StyleGAN and StyleGAN2.

Weakness

- Writing can be sigificantly improved. There are several typos and mistakes in writing (e.g., As the result, → As a result).

- Experiments with more datasets are needed because only two StyeGANs are used for evaluation. More SOTA GAN generated images need to be evaluated to provide the effectiveness and validity of this approach.

- Although they tackle an interesting problem, the proposed method is rather simple, and it is a combination of existing methods. Therefore, the novelty is rather weak.



**Summary Of The Paper:**

This work presents an approach to modify face memorability as a facial attribute using Generative Adversarial Networks (GANs). They essentially propose “memorability modification vector.” Their approach first determines the hyperplane and we can move the latent vector of each image, in the positive or negative direction of the normal vector of that hyperplane. They control the distance of the latent vector from the separating hyperplane to manipulate the memorability of that image. They conducted experiments with StyleGAN and StylyeGAN2 to demonstrate effectivness of their apporaching using FID and KID metrics.


**Summary Of The Review:**

Although this work tackles an interesting problem, the novelty is weak. Also, more experiments with different GAN datasets would be helpful. It is limited to StyleGANs. Also, the writing should be improved. Reproducibility is not certain as they did not provide the code.

---

> ### Author Response · Authors · 2021-11-22
> **Respond to Reviewer m2gg**
>
> Thank you for your thoughtful and accurate comments.
>
> * We English proofread our manuscript carefully and revised it to improve the writing.
>
> * In this work, our focus is on the analysis and modification of **face** image memorability. We use StyleGANs in our work because they are state-of-the-art models for generating **real-looking faces**. Our utilization of StyleGANs is required to create a dataset of realistic-looking faces. We use this generated dataset to find the memorability hyperplane in the model’s latent space. Not only that, for modifying the memorability of real faces, **we need StyleGANs to reconstruct real faces with high accuracy**.  To date, StyleGANs are the state-of-the-art models in reconstructing real-face images. Moreover, we want to analyze the relationship between memorability scores and facial attributes. We aim to investigate what attributes of a face contribute to the memorability score. StyleGANs provide an extended latent space which we leverage to derive a more accurate memorability hyperplane. Also, the face attributes of StyleGANs are especially disentangled in comparison to other GANs, which is required to accurately modify faces for memorability and study the attributes contributing to this. These are the reasons why we chose StyleGANs as the generator in our work. We clarified this in our revised manuscript. (page. 2, first paragraph)
>
> * In expanding our method and showing its validity and effectiveness, we added new experiments on the memorability of non-face objects in the appendix. We use StyleGAN2 independently pre-trained on cars, churches, horses, and cats to generate images in these categories. With our method, we are still able to modify the memorability scores of such objects. To show the effectiveness of our method, we mimic our previous experiments by generating 5k images in each category, modifying their memorability scores with different weights, and plotting the mean of their memorability scores.
>
> * To show that our method does not rely on the StyleGAN architecture, we added new experiments in the appendix where we applied our method on object images generated by BigGANs. We demonstrate the effectiveness of our method by testing it with different weights on 20k generated images by BigGANs.
>
> * We consider the simplicity of our method to be a strength. Currently, to the best of our knowledge, no study exists on modifying the memorability of real faces. Compared to [Goetschalckx et al., 2019], [Siarohin et al., 2017], our method works on modifying memorability of real images and it is much faster and more efficient in terms of memory and storage. Also, much less data is required to train the model. To summarize the novelties in our research:
>     + Ours is the first work that attempted to modify the memorability score of real faces and we were successful in doing so.
>     + We trained three new models for predicting the memorability score of the face images (our assessors). To the best of our knowledge, there is no publicly available model for predicting face memorability. We showed that these models work both on squared-shaped and oval-shaped faces.
>     + Although we were inspired by InterFaceGAN [Shen et al., 2020], there are some major differences. Firstly, InterFaceGAN is used for some binary attributes (e.g., age, gender, and wearing or not wearing eyeglasses) for which there is a binary classifier. However, in our work, we used the memorability scores of faces (which are continuous) and analyzed their distribution to divide them into two groups. In addition, **it is not trivial that there exists a separating hyperplane based on the memorability scores**, as the memorability score is a combination of the different face attributes.

---

> > ### Author Response · Authors · 2021-11-22
> > **Respond to Reviewer m2gg**
> >
> > * In continue to our last comment:
> >
> >     + Previous work (Isola et al. 2013; Khosla et al. 2015) showed the rank of the memorability scores for the images is highly consistent across participants, and time intervals. In other words, they showed that while the memorability values may change across time delays, the rank of memorability scores is highly preserved. Therefore, in the field of memorability, the rank of memorability scores are treated as time-independent, and observer-independent measures. As such, following previous works, we opted to use the scores from the images themselves to separate them and focused on the rank of the memorability scores. Hence, we can say that our method is robust to different assessors. However, Ganalyze uses the exact values of memorability scores which are not time and observer-independent and may cause some problems. Consider a case when there are two categories of images. When the range of memorability scores is low across one category and high in another, it will greatly affect the realness of the image if using the Ganaylze loss $E[(A(G(T(z,a))) - (A(G(z))-a))^2]$. Moreover, as the exact value of memorability is used, it cannot be used for square-shaped faces, because all the faces in the 10k us face database are oval-shaped and the assessor will be trained on oval-shaped faces and cannot predict the exact memorability scores of square-shaped faces very well. Although, it can predict the rank of the images.
> >     + By proposing our hyperplane method, we are able to modify the memorability score of the faces conditionally. Using the projection method explained in the paper, we can conditionally modify the memorability score while fixing selected attributes such as pose, smile, glasses, age, etc.

---

> ### Author Response · Authors · 2021-11-29
> **Response to Reviewer m2GG**
>
> We have added new parts to the paper and revised it based on your comments. We would like to hear your comments about them and appreciate it if you reconsider your decision.

---

### Decision · Program_Chairs · 2022-01-20

**Decision:**

Reject

**Comment:**

The reviewers raised a number of major concerns including the limited novelty of the proposed, inadequate motivation of the design choices and, most importantly, insufficient and unconvincing experimental evaluation presented. The authors’ rebuttal addressed some of the reviewers’ questions but failed to alleviate all reviewers’ concerns. Hence, I cannot suggest this paper for presentation at ICLR.